# Improved Algorithms for Online Submodular Maximization via First-order Regret Bounds

**Nicholas J. A. Harvey**
University of British Columbia
nickhar@cs.ubc.ca

**Christopher Liaw**
University of Toronto
cvliaw@cs.toronto.edu

**Tasuku Soma**
The University of Tokyo
tasuku_soma@mist.i.u-tokyo.ac.jp

## Abstract

We consider the problem of nonnegative submodular maximization in the online setting. At time step $t$, an algorithm selects a set $S_t \in \mathcal{C} \subseteq 2^V$ where $\mathcal{C}$ is a feasible family of sets. An adversary then reveals a submodular function $f_t$. The goal is to design an efficient algorithm for minimizing the expected approximate regret.

In this work, we give a general approach for improving regret bounds in online submodular maximization by exploiting *"first-order" regret bounds* for online linear optimization.

- For monotone submodular maximization subject to a matroid, we give an efficient algorithm which achieves a $(1 - c/e - \varepsilon)$-regret of $O(\sqrt{kT \ln(n/k)})$ where $n$ is the size of the ground set, $k$ is the rank of the matroid, $\varepsilon > 0$ is a constant, and $c$ is the average curvature. Even without assuming any curvature (i.e., taking $c = 1$), this regret bound improves on previous results of Streeter et al. (2009) and Golovin et al. (2014).
- For nonmonotone, unconstrained submodular functions, we give an algorithm with 1/2-regret $O(\sqrt{nT})$, improving on the results of Roughgarden and Wang (2018). Our approach is based on Blackwell approachability; in particular, we give a novel first-order regret bound for the Blackwell instances that arise in this setting.

## 1 Introduction

*Submodular maximization* is a ubiquitous optimization problem in machine learning, economics, and social networks [26]. A set function $f : 2^V \to \mathbb{R}$ on a ground set $V$ is *submodular* if it satisfies the *diminishing return property*: $f(X \cup \{i\}) - f(X) \geq f(Y \cup \{i\}) - f(Y)$ for $X \subseteq Y$ and $i \in V \setminus Y$. Given a nonnegative submodular function $f$ and a set family $\mathcal{C} \subseteq 2^V$, submodular maximization is the optimization problem $\max_{S \in \mathcal{C}} f(S)$. Although submodular maximization is NP-hard in general [11], approximation algorithms for various settings have been developed and they often perform very well in real-world applications [5, 6, 8, 12, 26, 31].

In this paper, we consider *online submodular maximization* in the full-information setting, which is formulated as the following repeated game between a player and an adversary. The player is given a set family $\mathcal{C}$ in a ground set $V$ in advance. For each round $t = 1, 2 \ldots$, the player plays a set $S_t \in \mathcal{C}$ possibly in a randomized manner and the adversary (perhaps knowing the player's strategy but not the randomized outcome) selects a submodular function $f_t : 2^V \to [0, 1]$. The player gains the reward

Table 1: A summary of our regret bounds and known bounds, where $n = |V|$, $k$ is the rank of the matroid, $c$ is the average curvature, $\varepsilon > 0$ is an arbitrary constant, and $T$ is the time horizon.

| setting | known results | **our results** |
|---|---|---|
| monotone+matroid $(\alpha = 1 - 1/e - \varepsilon)$ | $O(k\sqrt{nT})$ Golovin et al. [16] | $O(\sqrt{kT\ln(n/k)})$ Theorem 3.1 |
| monotone+matroid + bounded curvature $(\alpha = 1 - c/e - \varepsilon)$ | — | $O(\sqrt{kT\ln(n/k)})$ Theorem 3.1 |
| nonmonotone $(\alpha = 1/2)$ | $O(n\sqrt{T})$ Roughgarden and Wang [27] | $O(\sqrt{nT})$ Theorem 4.1 |
| monotone+cardinality $(\alpha = 1 - 1/e)$ | $O(\sqrt{kT\ln n})$ Streeter and Golovin [29] | $O(\sqrt{kT\ln(n/k)})$ Theorem 3.1 |

$f_t(S_t)$ and observes the submodular function $f_t$.[1] The performance is measured via the $\alpha$-regret:

$$\operatorname{Reg}_\alpha(T) := \alpha \max_{S^*} \sum_{t=1}^{T} f_t(S^*) - \sum_{t=1}^{T} f_t(S_t),$$

where $\alpha \in (0, 1]$ corresponds to the offline approximation ratio. The goal of online submodular maximization is to design an *efficient* algorithm for the player with a small $\alpha$-regret in expectation.

## 1.1 Our contribution

We provide efficient algorithms with improved regret bounds for various online submodular maximization. Our results are summarized in Table 1.

- For the case of monotone functions and a matroid constraint, (i.e., $f_t$ is nonnegative, monotone, and submodular, and $\mathcal{C}$ is a matroid), we provide an algorithm whose expected $(1 - c/e - \varepsilon)$-regret is at most $O(\sqrt{kT\ln(n/k)})$, where $n = |V|$, $k$ is the rank of the matroid $\mathcal{C}$, and $\varepsilon > 0$ is an arbitrary small constant. Here $c$ is the *curvature*[2] of $\sum_{t=1}^{T} f_t$. This result is the first $O(\sqrt{T})$ bound for the bounded curvature setting, generalizing the corresponding offline result [12, 31] to the online setting. In the case where $c = 1$, this result improves the best-known bound of $O(k\sqrt{nT})$ [16, 30] by a factor of $\tilde{\Omega}(\sqrt{kn})$. Note that the approximation ratio $1 - c/e$ is best possible for any algorithm making polynomially many queries to the objective function [31].
- For the nonmonotone and unconstrained setting (i.e., $f_t$ is nonnegative submodular and $\mathcal{C} = 2^V$), we devise an algorithm with $O(\sqrt{nT})$ expected $1/2$-regret, where $n = |V|$. This improves the best-known bound $O(n\sqrt{T})$ [27] by a factor of $\sqrt{n}$.

Finally, we remark that none of our algorithms require knowing the time horizon $T$ in advance.

## 1.2 Technical overview

The common ingredient of our algorithms is the use of *"first-order" regret bounds* for online linear optimization (OLO), which bound the regret of OLO algorithms in terms of the total gain or loss *of the best single action* rather than the time horizon $T$. We show that this data-dependent nature of first-order bounds enables us to exploit the structures of OLO subproblems appearing in online submodular maximization and it yields better bounds for approximate-regret. Below, we provide detailed description of this idea for each submodular maximization problem we study.

**Monotone**     Our algorithm is based on *online continuous greedy* [16, 30]. Roughly speaking, online continuous greedy reduces the problem to a series of OLO problems on a matroid polytope. For OLO on a matroid polytope, Golovin et al. [16] used *follow the perturbed leader (FPL)* [24], which gives

the $O(k\sqrt{nT})$ bound. The key observation to improving this bound is that the OLO subproblems that arise in this setting are structured in the sense that the sum of the rewards (across the subproblems) cannot be too large. Our technical contribution is a novel analysis of online continuous greedy showing that if one uses OLO algorithms with a first-order regret bound [25], then online continuous greedy yields the improved $O(\sqrt{kT\ln(n/k)})$ bound.

Furthermore, we show that combining the above techniques with the continuous greedy of Feldman [12] gives an algorithm for maximization of monotone submodular functions with bounded curvature under a matroid constraint. In particular, we show that the expected $(1 - c/e - \varepsilon)$-regret is bounded by $O(\sqrt{kT\ln(n/k)})$ where $c$ is the curvature of the *sum* of the submodular functions. We note that our algorithm does not require knowledge of $c$ beforehand.

**Nonmonotone**  At a high level, our algorithm for the nonmonotone case is similar to *online double greedy* of Roughgarden and Wang [27], which we will review briefly. They reduced the problem to a sequence of auxiliary online learning problems, called *USM balance subproblems*, for which they designed an algorithm with $O(\sqrt{T})$ regret. They also showed that if one has algorithms for the USM balance subproblems with regret $r_i$ for $i = 1, \ldots, n$, then online double greedy achieves $O(\sum_i r_i)$ regret bound, which gives the $O(n\sqrt{T})$ bound. Our contribution is a new algorithm for USM balance subproblems with a "first-order" regret bound. Combining this algorithm with a novel analysis of online double greedy, we obtain the improved $O(\sqrt{nT})$ bound. To design the first-order regret bound for USM balance subproblems, we exploit the *Blackwell approachability theorem* [1] and online dual averaging. Note that Roughgarden and Wang [27] did not use the Blackwell theorem and it is not obvious how to obtain a similar "first-order" bound from their analysis. We are not aware of other examples where similar regret bounds are known for Blackwell problems.

## 1.3  Related work

Online submodular maximization is a subfield of *online learning* [7]. A large body of work in online learning is devoted to *online convex optimization (OCO)*; see the monograph of Hazan [18]. Hazan and Kale [20] studied online submodular minimization through an OCO approach. The concept of first-order regret bounds originally appeared in Freund and Schapire [13] for the expert problem. We note that *second-order* regret bounds, where the range of the losses are not known and the regret depends on the square of the losses, have also been studied in the literature; see e.g. [19].

Studies of online submodular maximization were initiated by Streeter and Golovin [29]. They gave the first polynomial-time algorithm for the setting of monotone submodular functions and a cardinality constraint with $O(\sqrt{kT\ln n})$ expected $(1 - 1/e)$-regret, where $n = |V|$ is the size of the ground set and $k$ is the cardinality constraint constant. Subsequently, this result was generalized (with a slightly worse regret bound) to a partition matroid and a general matroid constraint in [16, 30]. For nonmonotone submodular maximization, Roughgarden and Wang [27] gave the first algorithm with $O(n\sqrt{T})$ expected $1/2$-regret. This was later generalized to nonmonotone $k$-submodular maximization by Soma [28] who gave an algorithm with $O(kn\sqrt{T})$ expected $1/2$-regret. Chen et al. [9, 10] and Zhang et al. [32] studied online *continuous submodular maximization* and obtained $O(\sqrt{T})$ approximate regret for various settings. Structurally, our algorithm for monotone submodular maximization bears some resemblance with those of Chen et al. [9, 10] in that we also use online linear optimization (OLO) algorithms as subroutines. However, the feedback for the OLO algorithms is different. In addition, we make use of the modular component of the submodular functions to obtain a result that depends on the curvature. Although the results of Chen et al. [9, 10] can be applied in our setting, they give a suboptimal dependence on $n$ and $k$. Zhang et al. [32] also study monotone submodular maximization subject to a matroid constraint in the "responsive bandit setting", where the algorithm can play and receive feedback for any set but receives a reward only for feasible sets. For this problem, they achieve expected $(1 - 1/e)$-regret at most $O(T^{8/9})$.

A series of studies developed black-box reductions of offline approximation algorithms to online no-approximate-regret algorithms [14, 21, 23]. These reductions apply only to linear functions.

## 1.4  Organization

The rest of the paper is organized as follows. Section 2 introduces some backgrounds of submodular maximization and online dual averaging. Section 3 presents our improved algorithm for monotone

functions of bounded curvature subject to a matroid constraint. Section 4 describes our algorithm for nonmonotone functions in the unconstrained setting.

## 2   Preliminaries

We denote the sets of real numbers, nonnegative real numbers, positive real numbers by $\mathbb{R}$, $\mathbb{R}_{\geq 0}$, and $\mathbb{R}_{>0}$, respectively. We also denote the set of nonpositive real numbers by $\mathbb{R}_{\leq 0}$. For a vector $c$, $|c|$ denotes the vector obtained by taking the element-wise absolute values.

Let $V$ be a finite ground set. For a set function $f : 2^V \to \mathbb{R}$, $S \subseteq V$, and $i \in V \setminus S$, we denote the marginal gain $f(S \cup \{i\}) - f(S)$ by $f(i \mid S)$. We sometimes abuse the notation for singletons, e.g., we denote $f(\{i\})$ by $f(i)$, $S \cup \{i\}$ by $S \cup i$, etc. For a vector $\ell \in \mathbb{R}^V$ and a subset $S \subseteq V$, we define $\ell(S) = \sum_{i \in S} \ell_i$. For a set function $f : 2^V \to \mathbb{R}$, its *multilinear extension* $F : [0, 1]^V \to \mathbb{R}$ is a smooth function defined as $F(x) = \mathbf{E}[f(R(x))] = \sum_{S \subseteq V} f(S) \prod_{i \in S} x_i \prod_{i \notin S} (1 - x_i)$, where $R(x)$ is a random set that independently contains each element $i \in V$ with probability $x_i$. It is well-known that $\nabla F \geq \mathbf{0}$ if $f$ is monotone and that $\frac{\partial F}{\partial x_i \partial x_j} \leq 0$ $(i \neq j)$ if $f$ is submodular [6].

A matroid is a set family $\mathcal{I} \subseteq 2^V$ such that (I1) $\emptyset \in \mathcal{I}$, (I2) $X \subseteq Y$ and $Y \in \mathcal{I}$ implies $X \in \mathcal{I}$, and (I3) $X, Y \in \mathcal{I}$ and $|X| < |Y|$ implies that there exists $i \in Y \setminus X$ such that $X \cup i \in \mathcal{I}$. The rank function of a matroid $\mathcal{M}$ is denoted by $\mathrm{rk}_{\mathcal{M}}$. The base polytope of a matroid $\mathcal{M}$ is a polytope defined as $B_{\mathcal{M}} = \{x \in \mathbb{R}^V_{\geq 0} : x(S) \leq \mathrm{rk}_{\mathcal{M}}(S) \ (S \subseteq V), \ x(V) = \mathrm{rk}_{\mathcal{M}}(V)\}$. *Rounding algorithms* take a vector $x$ in a base polytope $B_{\mathcal{M}}$ and output a random independent set $X \in \mathcal{I}$ such that $\mathbf{E}[f(X)] \geq F(x)$ for any monotone submodular function $f$ and its multilinear extension $F$. Examples of rounding algorithms include pipage rounding and swap rounding [6, 8].

**Online Linear Optimization and Online Dual Averaging.** Both of our algorithms make use of algorithms for online linear optimization (OLO) as a subroutine, which we will now briefly describe. Let $\mathcal{X} \subseteq \mathbb{R}^n$ be a convex set. In OLO, at each time step $t = 1, 2, \dots$ an algorithm chooses an element $x_t \in \mathcal{X}$ after which an adversary chooses a cost function $c_t \in [-1, 1]^n$. The goal is to minimize $\sum_{t=1}^T (c_t^\top x_t - c_t^\top z)$ for all $z \in \mathcal{X}$. One algorithm to achieve this is online dual averaging which is described in Appendix B. Here, we will just state the guarantee. For $x, y \in \mathbb{R}^n$, define KL-divergence $D_{\mathrm{KL}}(x, y) \coloneqq \sum_{i=1}^n x_i \ln \frac{x_i}{y_i} - x_i + y_i$. The following corollary is a restatement of Corollary B.3.

**Corollary 2.1.** *Let $x_1$ be an initial point and let $D \geq \max\{1, \sup_{u \in \mathcal{X}} D_{\mathrm{KL}}(u, x_1)\}$. Assuming that the cost vectors $c_t \in [-1, 1]^n$ then there is an algorithm for OLO that produces a sequence of iterates $x_1, x_2, \dots$ such that $\sum_{t=1}^T \left(c_t^\top x_t - c_t^\top z\right) \leq 3\sqrt{D}\sqrt{\sum_{t=1}^T |c_t|^\top x_t} + D$ for all $z \in \mathcal{X}$ and $T > 0$.*

Finally, $\Pi_{\mathcal{X}}^{\mathrm{KL}}(x) \coloneqq \operatorname{argmin}_{y \in \mathbb{R}} D_{\mathrm{KL}}(x, y)$ denotes the KL projection of $y$ onto the convex set $\mathcal{X}$.

## 3   Online monotone submodular maximization

Recall that the *curvature* of a monotone submodular function $f$ is defined as $c = 1 - \min_i \frac{f(i|V \setminus i)}{f(i)}$. Every monotone submodular function has curvature $c \in [0, 1]$ and linear functions have curvature $c = 0$. Our main result in this section is the following theorem.

**Theorem 3.1.** *For any constant $\varepsilon > 0$, there exists a polynomial-time algorithm for online submodular maximization subject to a matroid constraint whose expected $(1 - c/e - \varepsilon)$-regret is bounded by $O(\sqrt{kT \ln(n/k)})$ for every $T > 0$, where $n$ is the size of the ground set, $k$ is the rank of the matroid, and $c$ is the curvature of $\sum_{t=1}^T f_t$.*

Note that the curvature $c$ may change over time. This section gives an informal proof of Theorem 3.1 with a continuous-time algorithm; the discretized algorithm and analysis appears in Appendix E.

We will briefly mention the running time of the algorithm. The discretized algorithm makes $O\left(\frac{n^4}{\varepsilon^3} \log\left(\frac{n^3 T}{\varepsilon}\right)\right)$ calls to the evaluation oracle of the objective function $f_t$ and solves $O\left(\frac{n^3}{\varepsilon}\right)$ submodular function minimization problems in each round $t$. Note that submodular function minimization is used for the Bregman projection step (see Appendix D.3 in the supplementary materials).

### 3.1 Continuous-Time Algorithm

The main idea is to adapt the recent continuous greedy algorithm of Feldman [12] for maximizing a monotone submodular function. For a monotone submodular function $f$, we can define the corresponding modular function $\ell$ by

$$\ell(S) = \sum_{i \in S} f(i \mid V - i). \tag{3.1}$$

One can easily check that the set function $g := f - \ell$ is again monotone and submodular. The continuous-time version of the algorithm is presented in Algorithm 1.[3]

---

**Algorithm 1** Continuous-time algorithm

---

**Input:** Matroid $\mathcal{M}$ and dual averaging algorithms $\mathcal{A}_s$ on the base polytope $B_{\mathcal{M}}$ for $s \in [0, 1]$.
 1: Initialize dual averaging algorithms $\mathcal{A}_s$ for each $s \in [0, 1]$.
 2: **for** $t = 1, 2, \dots$ **do**
 3:     Set $x_t(0) = \mathbf{0}$.
 4:     **for** $s \in [0, 1]$ **do**
 5:         Move $x_t(s)$ via dynamics $\frac{\mathrm{d}x_t(s)}{\mathrm{d}s} = y_t(s)$, where $y_t(s) \in B_{\mathcal{M}}$ is the prediction from by $\mathcal{A}_s$.
 6:     Apply rounding to $x_t := x_t(1)$ and obtain $S_t$.
 7:     Play $S_t$ and observe $f_t$.
 8:     Compute the modular function $\ell_t$ for $f_t$ by (3.1) and let $g_t = f_t - \ell_t$.
 9:     **for** $s \in [0, 1]$ **do**
10:         Feedback cost vector $c_t = -e^{s-1}\nabla G_t(x_t(s)) - \ell_t$ to $\mathcal{A}_s$; $G_t$ is multilinear extension of $g_t$.

---

In Subsection 3.2, we will analyze Algorithm 1. In order to obtain a good regret bound on the problem, we require $\mathcal{A}_s$ (as defined in Algorithm 1) to have a first-order regret bound for which we can use Corollary 2.1. Finally, $\mathcal{A}_s$ requires performing a Bregman projection onto the matroid base polytope. The details of this can be found in Appendix D.3 in the supplementary materials.[4]

### 3.2 Analysis

Let $S^* \in \mathrm{argmax}_{S \in \mathcal{M}} \sum_{t=1}^{T} f_t(S)$ and let $r_s := \max_{z \in B_{\mathcal{M}}} \sum_{t=1}^{T} (e^{s-1}\nabla G_t(x_t(s)) + \ell_t)^{\top}(z - y_t(s))$ be the regret of $\mathcal{A}_s$ for $s \in [0, 1]$. The first lemma bounds the regret of Algorithm 1 in terms of $r_s$. The proof is similar to that in [12]; it can be found in Appendix D.1.

**Lemma 3.2.** *Let $S^* \in \mathrm{argmax}_{S \in \mathcal{M}} \sum_{t=1}^{T} f_t(S)$. Then Algorithm 1 outputs $S_1, \dots, S_T$ such that* $\mathbf{E}[(1 - c/e) \sum_{t=1}^{T} f_t(S^*) - \sum_{t=1}^{T} f_t(S_t)] \leq R$, *where* $R = \int_0^1 r_s \mathrm{d}s$.

It remains to bound $R$. Let $\rho_s := \sum_{t=1}^{T} (e^{s-1}\nabla G_t(x_t(s)) + \ell_t)^{\top} y_t(s)$ be the reward received by algorithm $\mathcal{A}_s$. Suppose each $\mathcal{A}_s$ is an instance of the algorithm promised by Corollary 2.1 with initial point $y_1(s) = \Pi_{\mathcal{B}_{\mathcal{M}}}^{\mathrm{KL}}\left(\frac{k}{n}\mathbf{1}\right)$. By standard properties (Fact A.4 and Fact A.5), we have $\sup_{u \in \mathcal{X}} D_{\mathrm{KL}}(u, x_1) \leq k\ln(n/k)$. Applying Corollary 2.1 (with $c_t = -e^{s-1}\nabla G_t(y_t(s)) - \ell_t \in \mathbb{R}_{\leq 0}^n$ and $D = k\ln(n/k)$), we have $r_s \leq 3\sqrt{k\ln(n/k)}\sqrt{\rho_s} + k\ln(n/k)$.

**Lemma 3.3.** *Suppose that* $r_s \leq 3\sqrt{k\ln(n/k)}\sqrt{\rho_s} + k\ln(n/k)$. *Then* $R \leq 4\sqrt{k\ln(n/k)}\sqrt{T}$.

We will need a claim to bound $\int_0^1 \rho_s \, \mathrm{d}s$; we relegate the proof to Appendix D.2.

**Claim 3.4.** $\int_0^1 \rho_s \, \mathrm{d}s \leq T$.

*Proof of Lemma 3.3.* If $T \leq k\ln(n/k)$ then we trivially bound $r_s \leq T \leq \sqrt{k\ln(n/k)}\sqrt{T}$. Since $R = \int_0^1 r_s \, \mathrm{d}s$, we have $R \leq \sqrt{k\ln(n/k)}\sqrt{T}$. Henceforth, we assume $T \geq k\ln(n/k)$. We have that $R = \int_0^1 r_s \, \mathrm{d}s \leq 3\sqrt{k\ln(n/k)} \int_0^1 \sqrt{\rho_s} \, \mathrm{d}s + k\ln(n/k)$ by the hypothesis of the lemma. By

Jensen's Inequality, we have $\int_0^1 \sqrt{\rho_s}\,\mathrm{d}s \le \sqrt{\int_0^1 \rho_s\,\mathrm{d}s} \le \sqrt{T}$ where the last inequality is by Claim 3.4. Finally, as $k\ln(n/k) \le \sqrt{Tk\ln(n/k)}$, we conclude that $R \le 4\sqrt{k\ln(n/k)} \cdot \sqrt{T}$. $\qquad\square$

# 4 Online nonmonotone submodular maximization

In this section, we prove the following theorem.

**Theorem 4.1.** *For online nonmonotone submodular maximization, there exists a polynomial time algorithm whose expected $1/2$-regret is $O(\sqrt{nT})$ for every $T > 0$, where $n = |V|$.*

In Subsection 4.1, we review the online double greedy algorithm by [27] and introduce USM-balance subproblems. Subsection 4.2 describes the necessary background of Blackwell approachability. In Subsection 4.3, we prove our main technical result, a first-order regret bound for Blackwell instances arising from USM-balance subproblems. Given the first-order regret bound, the proof of Theorem 4.1 is fairly straightforward and deferred to Appendix F.1 (due to space constraints).

## 4.1 Online double greedy and USM-balance subproblem

First, we review the online double greedy algorithm by [27]. Their algorithm is based on the well-known double greedy algorithm [5]. At the beginning of each time $t$, the algorithm initializes sets $X_t = \emptyset$ and $Y_t = [n]$. For each element $i$, the algorithm updates $X_t$ and $Y_t$ using a probability vector $p_{t,i} = (p_{t,i}^+, p_{t,i}^-) \in \mathbb{R}^2$. The pseudo code is given in Algorithm 2. In terms of time complexity, in each iteration $t$, the algorithm makes a single call to each of the $n$ USM-balance game algorithms and $O(n)$ calls to the submodular function $f_t$. Note that each of the USM-balance algorithms may be executed in $O(1)$ time because the underlying convex optimization problem is $O(1)$-dimensional.

---

**Algorithm 2** Online Double Greedy

1: Set up USM-balance subproblem algorithms $\mathcal{A}_i$ for $i = 1, \dots, n$.
2: **for** $t = 1, 2, \dots$ **do**
3:      Initialize $X_{t,0} = \emptyset$ and $Y_{t,0} = [n]$.
4:      **for** $i = 1, \dots, n$ **do**
5:          Call the USM-balancing game algorithm $\mathcal{A}_i$ to obtain $p_{t,i} = (p_{t,i}^+, p_{t,i}^-)$.
6:          With probability $p_{t,i}^+$, update $X_{t,i} = X_{t,i-1} \cup i$ and $Y_{t,i} = Y_{t,i-1}$. Otherwise, update $X_{t,i} = X_{t,i-1}$ and $Y_{t,i} = Y_{t,i-1} \setminus i$.
7:      **return** $S_t := X_{t,n}$
8:      **for** $i = 1, \dots, n$ **do**
9:          Feedback $\Delta_{t,i} = (f_t(X_{t,i-1} \cup i) - f_t(X_{t,i-1}), f_t(Y_{t,i-1} \setminus i) - f_t(Y_{t,i-1}))$ to $\mathcal{A}_i$.

---

The approximation ratio of the algorithm crucially depends on the choice of $p_{t,i}$. In the offline setting [5], the following choice is known to give a $1/2$-approximation:

$$(p_{t,i}^+, p_{t,i}^-) = \begin{cases} (0,1) & \text{if } \Delta_{t,i}^+ \le 0 \\ (1,0) & \text{if } \Delta_{t,i}^- < 0 \\ \left( \frac{\Delta_{t,i}^+}{\Delta_{t,i}^+ + \Delta_{t,i}^-}, \frac{\Delta_{t,i}^-}{\Delta_{t,i}^+ + \Delta_{t,i}^-} \right) & \text{otherwise} \end{cases},$$

where $\Delta_{t,i}^+ := f_t(X_{t,i-1} \cup i) - f_t(X_{t,i-1})$ and $\Delta_{t,i}^- := f_t(Y_{t,i-1} \setminus i) - f_t(Y_{t,i-1})$. We note that $\Delta_{t,i}^+ + \Delta_{t,i}^- \ge 0$ [5, Lemma 2.1]. The key ingredient of Roughgarden and Wang [27] is predicting $p_{t,i}$ in an online fashion by considering another online learning problem, a *USM-balance subproblem*.

**Definition 4.2** (USM-balance subproblem [27])**.** *An instance of USM-balance subproblems is the following repeated game: For $t = 1, 2, \dots$,*

- *A player plays a two dimensional probability vector $p_t = (p_t^+, p_t^-)$.*

- *An adversary plays a vector $\Delta_t = (\Delta_t^+, \Delta_t^-) \in [-1, 1]^2$ such that $\Delta_t^+ + \Delta_t^- \ge 0$.*

*The regret of the USM-balance subproblem is defined as*

$$r(T) := \max \left\{ \sum_{t=1}^T p_t^- \Delta_t^+, \sum_{t=1}^T p_t^+ \Delta_t^- \right\} - \frac{1}{2} \sum_{t=1}^T \left( p_t^+ \Delta_t^+ + p_t^- \Delta_t^- \right). \tag{4.1}$$

In the offline setting Eq. (4.1) is used as an upper bound on a certain potential function (see [5, Proof of Lemma 3.1]). Indeed, choosing $p_t$ such that Eq. (4.1) is non-positive immediately gives a $1/2$-approximation in the offline setting. Lemma 4.3 extends this observation and relates the regret of USM-balance games with the $1/2$-regret of Online Double Greedy.

**Lemma 4.3** (Roughgarden and Wang [27, Theorem 2.1]). *Suppose that the USM-balance subproblem algorithms $\mathcal{A}_i$ have regret $r_i(T)$ for $i \in [n]$. Then, Online Double Greedy outputs $S_t$ such that*

$$\mathbf{E}\left[\frac{1}{2} \max_{S^*} \sum_{t=1}^{T} f_t(S^*) - \sum_{t=1}^{T} f_t(S_t)\right] \leq \sum_{i=1}^{n} \mathbf{E}[r_i(T)]. \tag{4.2}$$

It suffices to show that the USM-balance subproblem can be solved with small expected regret. In [27], they design an efficient algorithm for the USM-balance subproblem with $O(\sqrt{T})$ regret. However, their algorithm was cleverly hand-crafted for the USM-balance subproblem. As mentioned in a footnote in [27], it is possible to design an algorithm via Blackwell approachability [1, 3].

Note that the $O(\sqrt{T})$ bound on the USM-balance subproblem is a worst-case zeroth-order regret bound. Suppose instead that we had a *first-order* regret bound, say (for example), $r_i(T) \lesssim \sqrt{\sum_t p_{t,i}^+ \Delta_{t,i}^+ + p_{t,i}^- \Delta_{t,i}^-}$ (the index $i$ corresponds to $\mathcal{A}_i$ in Line 5). The quantity $\mathbf{E}[\sum_i \sum_t p_{t,i}^+ \Delta_{t,i}^+ + p_{t,i}^- \Delta_{t,i}^-]$ is the expected reward for Online Double Greedy and is at most $T$. Hence, $\sum_t p_{t,i}^+ \Delta_{t,i}^+ + p_{t,i}^- \Delta_{t,i}^-$ cannot be $\Theta(T)$ for all $i$; consequently $r_i(T)$ cannot all be large. Although this "first-order" bound does not hold, because the quantity in the square-root can be negative, one can formalize this observation as in the following lemma, which suffices to show the desired $O(\sqrt{nT})$ bound.

**Lemma 4.4.** *There exists an efficient algorithm $\mathcal{A}$ for the USM-balance subproblem such that for some sets $C^+, C^- \subseteq \mathbb{N}$,*

$$r(T) \leq O\left(\max\left\{\sqrt{\sum_{t=1}^{T} p_t^- |\Delta_t^+|}, \sqrt{\sum_{t=1}^{T} p_t^+ |\Delta_t^-|}\right\} + \sqrt{\sum_{t \in C^+ \cap [T]} \alpha_t} + \sqrt{\sum_{t \in C^- \cap [T]} \beta_t} + 1\right). \tag{4.3}$$

*Here, $\alpha_t = \frac{3}{2} p_t^+ \Delta_t^+ + \frac{1}{2} p_t^- \Delta_t^-$ and $\beta_t = \frac{1}{2} p_t^+ \Delta_t^+ + \frac{3}{2} p_t^- \Delta_t^-$. Moreover,*

- *the events $t \in C^+$, $t \in C^-$ depend only on $p_t, \Delta_t$; and*
- *$\alpha_t \geq 0$ for all $t \in C^+$ and $\beta_t \geq 0$ for all $t \in C^-$.*

At this point, the proof of Theorem 4.1 follows from Lemma 4.4 via some calculations which we defer to Appendix F.1 in the supplementary material. We stress that the important point is that the bound in Lemma 4.4 depends on the actual sequence of inputs the algorithm receives. Next, we will prove Lemma 4.4 by opening up the reduction of Blackwell approachability to OLO and show that, with an appropriate OLO algorithm, one can obtain a first-order regret bound for Blackwell approachability in the setting of USM-balance subproblems.

## 4.2 Blackwell approachability

**Definition 4.5** (Blackwell instance). *A Blackwell instance is a tuple $(\mathcal{X}, \mathcal{Y}, u, \mathcal{S})$, where $\mathcal{X} \subseteq \mathbb{R}^n$, $\mathcal{Y} \subseteq \mathbb{R}^m$, $\mathcal{S} \subseteq \mathbb{R}^d$ are closed convex sets and $u : \mathcal{X} \times \mathcal{Y} \to \mathbb{R}^d$ is a biaffine function, i.e., $u(x, \cdot)$ is affine for any $x \in \mathcal{X}$ and vise versa. An instance is said to be*

- satisfiable *if $\exists x \in \mathcal{X} \, \forall y \in \mathcal{Y}$ such that $u(x, y) \in \mathcal{S}$.*
- response-satisfiable *if $\forall y \in \mathcal{Y} \, \exists x \in \mathcal{X}$ such that $u(x, y) \in \mathcal{S}$.*
- halfspace-satisfiable *if any halfspace $H$ containing $\mathcal{S}$ is satisfiable.*
- approachable *if there exists an algorithm $\mathcal{A}$ such that for any $(y_t) \subseteq \mathcal{Y}$, the sequence $x_t = \mathcal{A}(y_1, \ldots, y_{t-1})$ satisfies $\mathrm{dist}(\frac{1}{T} \sum_{t=1}^{T} u(x_t, y_t), \mathcal{S}) \to 0$ as $T \to \infty$.*

In the above definition and the rest of the paper, we use $\mathrm{dist}(x, \mathcal{S}) := \inf_{y \in \mathcal{S}} \|x - y\|_2$ to denote the Euclidean distance from the point $x$ to $\mathcal{S}$.

**Theorem 4.6** (Blackwell Approachability Theorem [3]). *Let $\mathcal{B} = (\mathcal{X}, \mathcal{Y}, u, \mathcal{S})$ be a Blackwell instance. Then $\mathcal{B}$ is approachable if and only if $\mathcal{B}$ is response-satisfiable if and only if $\mathcal{B}$ is halfspace-satisfiable.*

Abernethy et al. [1] gave an algorithmic version of the Blackwell theorem via its connection to online linear optimization. A key ingredient of the algorithm is the concept of a *halfspace oracle*.

**Definition 4.7** (Halfspace oracle). *A halfspace oracle is an oracle that finds $x \in \mathcal{X}$ for given a halfspace $H \supseteq \mathcal{S}$ such that $u(x, y) \in H$ for all $y \in \mathcal{Y}$.*

They showed that given a halfspace oracle and an OLO algorithm on a certain convex set defined from an Blackwell instance, one can construct an efficient algorithm to produce an approaching sequence.

The USM-balancing subproblem can be cast as a Blackwell instance as follows. Let $\mathcal{X} = \{p = (p^+, p^-) \in [0,1]^2 : p^+ + p^- = 1\}$, $\mathcal{Y} = \{\Delta = (\Delta^+, \Delta^-) \in [-1,1]^2 : \Delta^+ + \Delta^- \geq 0\}$, and

$$u(p, \Delta) = \begin{bmatrix} p^- \cdot \Delta^+ - 1/2 p^\top \Delta \\ p^+ \cdot \Delta^- - 1/2 p^\top \Delta \end{bmatrix}, \qquad \mathcal{S} = \mathbb{R}_{\leq 0}^2. \tag{4.4}$$

Note that this instance is response-satisfiable, since if we know $\Delta$, one can set $p$ as in offline double greedy. By Theorem 4.6, there exists an algorithm $\mathcal{A}$ for producing an approaching sequence $p_t$. This yields a regret guarantee in the USM-balance subproblem because (recall Eq. (4.1)) $\frac{1}{T} \cdot r(T) \leq \text{dist}\left(\frac{1}{T} \sum_{t=1}^{T} u(p_t, \Delta_t), \mathcal{S}\right)$. In addition, one can construct an efficient halfspace oracle for this Blackwell instance via a standard LP duality argument; the details can be found in Appendix F.

### 4.3 First-order regret bound for Blackwell and proof of Lemma 4.4

In this section, we prove Lemma 4.4 via a reduction to the Blackwell approachability. Algorithm 3 shows the reduction from Blackwell approachability to online linear optimization.[5] Lemma 4.8 formalizes the relationship between no-regret learning and Blackwell approachability. In this section, let $\text{Reg}_{\mathcal{A}}(T)$ denote the regret of the dual averaging algorithm $\mathcal{A}$.

---

**Algorithm 3** An improved algorithm for USM-balance subproblems

---

**Input:** Halfspace oracle for Blackwell instance and $K = \mathcal{S}^\circ \cap B_2(1) = \{z \in \mathbb{R}_{\geq 0}^2 : \|z\|_2 \leq 1\}$.
1: Initialize dual averaging algorithm $\mathcal{A}$ with feasible set $K$, initial point $x_1 = (1/\sqrt{2}, 1/\sqrt{2})$, negative entropy mirror map $\Phi(z) := \sum_i z_i \ln(z_i)$.
2: **for** $t = 1, 2, \ldots$ **do**
3: $\quad x_t \leftarrow \mathcal{A}(c_1, \ldots, c_{t-1})$
4: $\quad$ Call the halfspace oracle for a halfspace $H = \{z : x_t^\top z \leq 0\}$ to obtain $p_t$.
5: $\quad$ Play $p_t$ and observe $\Delta_t$.
6: $\quad$ Set cost $c_t \leftarrow -u(p_t, \Delta_t)$.

---

**Lemma 4.8** (Abernethy et al. [1, Theorem 17]). *The output of Algorithm 3 satisfies $\frac{1}{T} r(T) \leq \text{dist}\left(\frac{1}{T} \sum_{t=1}^{T} u(p_t, \Delta_t), \mathcal{S}\right) \leq \frac{1}{T} \text{Reg}_{\mathcal{A}}(T)$.*

Corollary 2.1 asserts that online dual averaging algorithm with appropriate step sizes guarantees[6]

$$\text{Reg}_{\mathcal{A}}(T) \leq 6\sqrt{D} \sqrt{\sum_{t=1}^{T} |c_t|^\top x_t + 2D}, \tag{4.5}$$

where $D := \max\{1, \max_{z \in K} D_{\text{KL}}(z, x_1)\}$. Next, we claim $D = O(1)$ (proof in Appendix F).

**Claim 4.9.** $D_{\text{KL}}(x, x_1) \leq 2$ *for all $x \in B_2(1) \cap \mathbb{R}_{\geq 0}$.*

It now suffices to bound $\sum_{t \leq T} |c_t|^\top x_t$. Recall that $c_t = -u(p_t, \Delta_t)$ (defined in Eq. (4.4)), i.e.

$$c_t^+ = \frac{1}{2}(p_t^+ \cdot \Delta_t^+ + p_t^- \cdot \Delta_t^-) - p_t^+ \cdot \Delta_t^- \quad \text{and} \quad c_t^- = \frac{1}{2}(p_t^+ \cdot \Delta_t^+ + p_t^- \cdot \Delta_t^-) - p_t^- \cdot \Delta_t^+.$$

We define $\alpha_t = \frac{3}{2}p_t^+ \Delta_t^+ + \frac{1}{2}p_t^- \Delta_t^-$ and $\beta_t = \frac{1}{2}p_t^+ \Delta_t^+ + \frac{3}{2}p_t^- \Delta_t^-$. Finally, define $C^+ = \{t : c_t^+ \geq 0\}$ and $C^- = \{t : c_t^- \geq 0\}$. The proof of the following lemma can be found in Appendix F.

**Lemma 4.10.** *In the setting of the USM-balance subproblem, we have*

$$\sum_{t \leq T} |c_t|^\top x_t \leq \sum_{t \in C^+ \cap [T]} \alpha_t + \sum_{t \in C^- \cap [T]} \beta_t + \sum_{t \leq T} 2(p_t^+ |\Delta_t^-| + p_t^- |\Delta_t^+|).$$

*Proof of Lemma 4.4.* From Eq. (4.5), using Claim 4.9, Lemma 4.10, and $\sqrt{a+b} \leq \sqrt{a} + \sqrt{b}$ gives

$$\text{Reg}_{\mathcal{A}}(T) \leq O\left(\sqrt{\sum_{t \in C^+ \cap [T]} \alpha_t} + \sqrt{\sum_{t \in C^- \cap [T]} \beta_t} + \sqrt{\sum_{t \leq T} p_t^+ |\Delta_t^-| + p_t^- |\Delta_t^+|} + 1\right),$$

Finally, we bound $\sum_{t \leq T} p_t^+ |\Delta_t^-| + p_t^- |\Delta_t^+| \leq 2 \max\left\{\sum_{t \leq T} p_t^+ |\Delta_t^-|, \sum_{t \leq T} p_t^- |\Delta_t^+|\right\}$ to obtain the bound as written in the lemma. $\qquad\square$

## Broader Impact

This is a theoretical work and does not present any foreseeable societal consequences.

## Acknowledgments and Disclosure of Funding

The authors thank the anonymous referees for their useful comments. N.H. was supported by Canada Research Chairs Program and an NSERC Discovery Grant. C.L. was supported by an NSERC graduate scholarship. T.S. was supported by the Operations Research Society of Japan and JST, ERATO, Grant Number JPMJER1903, Japan.

## Footnotes

[1]Formally, each submodular function $f_t$ is given as a value oracle to the player after $S_t$ is chosen.

[2]The curvature $c$ of a nonnegative monotone submodular function $f$ is defined as $c = 1 - \min_{i \in V} \frac{f(V \setminus \{i\})}{f(\{i\})}$.

[3]Note that in Algorithm 1, we assume the OLO algorithm $\mathcal{A}_s$ is trying to minimize *losses*; since we care about rewards, we negate the reward vectors to get cost vectors.

[4] Missing proofs and appendices can be found in the supplementary materials.

[5] Note that the OLO algorithm requires a KL projection onto $K := B_2(1) \cap \mathbb{R}_{\geq 0}^2$. This is a two-dimensional convex minimization problem which can be easily solved up to any desired accuracy; the details are omitted from this version of the paper.

[6] The factor of 2 difference between this bound and Corollary 2.1 is because $c_t \in [-2, 2]^2$ in this setting.

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
