[Supplementary Material]

# A   Standard facts

**Fact A.1.** $e^x \leq 1 + x + x^2$ *for all* $x \in (-\infty, 1]$.

**Claim A.2.** *Suppose that $x$ satisfies*

$$x^2 - \alpha x - \beta \leq 0,$$

*where $\alpha, \beta \geq 0$ are constants. Then*

$$x \leq \frac{\alpha + \sqrt{\alpha^2 + 4\beta}}{2}.$$

**Claim A.3.** *Let $f$ be a non-negative submodular function on $[n]$ that is bounded above by $1$. Let $X_0, \ldots, X_s$ be a monotone sequence of sets, i.e. either $X_0 \subseteq \ldots \subseteq X_s \subseteq [n]$ or $[n] \supseteq X_0 \supseteq \ldots \supseteq X_s$. Then for any $I \subseteq [s]$,*

$$\sum_{i \in I} f(X_i) - f(X_{i-1}) \leq 1.$$

*Proof.* First, suppose that $X_i$ are monotone increasing. Construct a sequence $X'_i$ as follows. Set $X'_0 = X_0$. If $i \notin I$ then set $X'_i = X'_{i-1}$. If $i \in I$ then set $X'_i = X'_{i-1} \cup (X_i \setminus X_{i-1})$. In this case,

$$\sum_{i \in I} f(X_i) - f(X_{i-1}) \leq \sum_{i \in I} f(X'_i) - f(X'_{i-1}) = \sum_{i=1}^{s} f(X'_i) - f(X'_{i-1}) = f(X'_s) - f(X'_0) \leq 1.$$

For the monotone decreasing case, consider the submodular function $g(X) = f([n] - X)$ and set $Y_i = [n] - X_i$. Observe that $Y_i$ are monotone increasing sets. Then

$$\sum_{i \in I} f(X_i) - f(X_{i-1}) = \sum_{i \in I} g([n] - X_i) - g([n] - X_{i-1}) = \sum_{i \in I} g(Y_i) - g(Y_{i-1}) \leq 1,$$

where the last inequality is by the monotone increasing case. $\qquad\square$

**Fact A.4.** *Suppose that $p \in [0,1]^n$ satisfies $\sum_i p_i = k$. Let $q = \frac{k}{n}\mathbf{1}$. Then $D_{\mathrm{KL}}(p,q) \leq k\ln(n/k)$.*

*Proof.* We have $D_{\mathrm{KL}}(p,q) = \sum_i p_i \ln \frac{p_i}{k/n} = \sum_i p_i \ln(n/k) + \sum_i p_i \log p_i \leq k\ln(n/k)$, where in the last inequality we used $\sum_i p_i \ln p_i \leq 0$. $\qquad\square$

**Fact A.5.** *Let $\pi = \Pi_{\mathcal{X} \cap \mathcal{D}}^{\Phi}(y)$. Then $D_{\Phi}(x, \pi) \leq D_{\Phi}(x, y)$ for all $x \in \mathcal{X} \cap \mathcal{D}$.*

**Proposition A.6.** *Let $u > 0$ and $a_1, \ldots, a_T \in [0, u]$. Then*

$$\sum_{t=1}^{T} \frac{a_t}{\sqrt{u + \sum_{i<t} a_i}} \leq 2\sqrt{\sum_{t=1}^{T} a_t}.$$

*Proof.* This follows from [2, Lemma 3.5]. $\qquad\square$

# B   Online Dual Averaging

Both of our algorithms make use of the online dual averaging algorithm, which we will briefly describe here (see Bubeck [4, Chapter 4] for a more detailed exposition). Let $\mathcal{D} \subseteq \mathbb{R}^n$ be an open convex set and $\Phi \colon \mathcal{D} \to \mathbb{R}$ be a strictly convex and differentiable function on $\mathcal{D}$. The function $\Phi$ is called the *mirror map*. We further require that $\nabla\Phi(\mathcal{D}) = \mathbb{R}^n$ and that $\lim_{x \to \partial\mathcal{D}} \|\nabla\Phi(x)\| = +\infty$. Let $\mathcal{X}$ denote the feasible region, which is assumed to be closed and convex. Moreover, $\mathcal{X} \subseteq \overline{\mathcal{D}}$ and $\mathcal{X} \cap \mathcal{D} \neq \emptyset$. Finally, $D_{\Phi}(x, y) := \Phi(x) - \Phi(y) - \langle \nabla\Phi(y), x - y \rangle$ is the Bregman divergence of $\Phi$. We use the notation $\Pi_{\mathcal{X} \cap \mathcal{D}}^{\Phi}(y) = \mathrm{argmin}_{x \in \mathcal{X} \cap \mathcal{D}} D_{\Phi}(x, y)$ to denote the Bregman projection of $y$ onto $\mathcal{X}$ with $\Phi$ as the mirror map.

The gradient of the mirror map $\nabla\Phi \colon \mathcal{D} \to \mathbb{R}^n$ and the gradient of its conjugate $\nabla\Phi^* \colon \mathbb{R}^n \to \mathcal{D}$ are mutually inverse bijections between the primal space $\mathcal{D}$ and the dual space $\mathbb{R}^n$. We will adopt the

following notational convention. Any vector in the primal space will be written without a hat, such as $x \in \mathcal{D}$. The same letter with a hat, namely $\hat{x}$, will denote the corresponding dual vector:

$$\hat{x} := \nabla\Phi(x) \qquad \text{and} \qquad x := \nabla\Phi^*(\hat{x}) \qquad \text{for all letters } x.$$

In our applications, we take $\mathcal{D} = \mathbb{R}_{>0}^n$ and $\Phi(x) = \sum_i x_i \ln x_i$. In Section 3, we take $\mathcal{X}$ to be the matroid base polytope while in Section 4 we take $\mathcal{X}$ to be the unit Euclidean ball intersected with the positive orthant. In this case,

$$\nabla\Phi(x)_i = \ln(x_i) + 1 \qquad \text{and} \qquad \nabla\Phi^*(\hat{x})_i = \exp(\hat{x}_i - 1) \tag{B.1}$$

and the Bregman divergence is the generalized KL divergence, i.e.

$$D_\Phi(x, y) = D_{\text{KL}}(x, y) = \sum_{i=1}^n x_i \ln \frac{x_i}{y_i} - x_i + y_i.$$

We note that the assumptions required above on $\mathcal{D}, \Phi, \mathcal{X}$ are satisfied with these choices. Algorithm 4 describes the online dual averaging algorithm. In the entirety of this section, we will always assume that $f_t$ denote convex functions and that $\|f_t\|_\infty \le 1$ for all $t$.

---

**Algorithm 4** Online Dual Averaging

**Input:** Initial point $x_1 \in \mathcal{X} \cap \mathcal{D}$, mirror map $\Phi$, and learning rate $\eta : \mathbb{N} \to \mathbb{R}_{>0}$.
1: $\hat{x}_1 \leftarrow \nabla\Phi(x_1)$
2: **for** $t = 1, 2, \ldots,$ **do**
3:     Play $x_t$, incur cost $f_t(x_t)$, and receive subgradient $\hat{g}_t \in \partial f_t(x_t)$.
4:     $\hat{y}_{t+1} \leftarrow \hat{x}_1 - \eta_{t+1} \sum_{i \le t} \hat{g}_i$
5:     $y_{t+1} \leftarrow \nabla\Phi^*(\hat{y}_{t+1})$
6:     $x_{t+1} \leftarrow \Pi_{\mathcal{X} \cap \mathcal{D}}^\Phi(y_{t+1})$

---

The following is a standard, but quite general, analysis of the online dual averaging algorithm.

**Theorem B.1.** *Assume that $\eta_t \ge \eta_{t+1} > 0$ for all $t \ge 1$. Let $\{x_t\}_{t \ge 1}$ be the sequence of iterates generated by Algorithm 4. Let $v_t = \nabla\Phi^*(\hat{x}_t - \eta_t \hat{g}_t)$. Then for any mirror map $\Phi$, any sequence of convex functions $\{f_t\}_{t \ge 1}$ with each $f_t : \mathcal{X} \to \mathbb{R}$, and any $z \in \mathcal{X}$,*

$$\sum_{t=1}^T \left( f_t(x_t) - f_t(z) \right) \le \sum_{t=1}^T \frac{D_\Phi(x_t, v_t)}{\eta_t} + \frac{\sup_{u \in \mathcal{X}} D_\Phi(u, x_1)}{\eta_{T+1}} \qquad \forall T > 0. \tag{B.2}$$

If the cost functions are linear, say $f_t(x) = c_t^\top x$, and the mirror map is $\Phi(x) = \sum_i x_i \ln x_i$ then we have the following bound on the regret.

**Corollary B.2.** *Assume that $\eta_1 \le 1$ and $\eta_t \ge \eta_{t+1} > 0$ for all $t \ge 1$. Assume that $\Phi(x) = \sum_i x_i \ln x_i$. Let $\{x_t\}_{t \ge 1}$ be the sequence of iterates generated by Algorithm 4. Then for any sequence of cost vectors $c_t \in [-1, 1]^n$ and any $z \in \mathcal{X}$,*

$$\sum_{t=1}^T \left( c_t^\top x_t - c_t^\top z \right) \le \sum_{t=1}^T \eta_t |c_t|^\top x_t + \frac{\sup_{u \in \mathcal{X}} D_{\text{KL}}(u, x_1)}{\eta_{T+1}} \qquad \forall T > 0.$$

**Corollary B.3.** *In the setting of Corollary B.2, if we take $\eta_t = \sqrt{\frac{D}{D + \sum_{j < t} |c_t|^\top x_t}}$, where $D \ge \max\{1, \sup_{u \in \mathcal{X}} D_{\text{KL}}(u, x_1)\}$ then for any $z \in \mathcal{X}$,*

$$\sum_{t=1}^T \left( c_t^\top x_t - c_t^\top z \right) \le 3\sqrt{D} \sqrt{\sum_{t=1}^T |c_t|^\top x_t} + D.$$

The proofs of the previous two corollaries are in Appendix C.

## C  Proofs from Appendix B

*Proof of Corollary B.2.*  Each term in the sum of (B.2) may be bounded as follows:

$$
\begin{aligned}
\frac{D_{\mathrm{KL}}\left(x_t, v_t\right)}{\eta_t} &= \frac{1}{\eta_t} \sum_{i=1}^{n} \left( x_{t,i} \ln \frac{x_{t,i}}{v_{t,i}} - x_{t,i} + v_{t,i} \right) \\
&= \frac{1}{\eta_t} \sum_{i=1}^{n} x_{t,i} \left( -\eta_t c_{t,i} - 1 + e^{\eta_t c_{t,i}} \right) \\
&\leq \frac{1}{\eta_t} \sum_{i=1}^{n} x_{t,i} \left( -\eta_t c_{t,i} - 1 + (1 + \eta_t c_{t,i} + \eta_t^2 c_{t,i}^2) \right) \qquad \text{(by Fact A.1)} \\
&= \eta_t \sum_{i=1}^{n} x_{t,i} c_{t,i}^2 \ \leq\ \eta_t \sum_{i=1}^{n} x_{t,i} |c_{t,i}| = \eta_t |c_t|^\top x_t.
\end{aligned}
$$

In the first equality we used that $\eta_t c_{t,i} \leq 1$ and in the last equality we used that $c_{t,i} \in [-1, 1]$. $\qquad\square$

*Proof of Corollary B.3.*  Note first that $\eta_t$ is a decreasing sequence and $\eta_1 \leq 1$. By Corollary B.2, we bound

$$
\sum_{t=1}^{T} \left( c_t^\top x_t - c_t^\top z \right) \leq \sum_{t=1}^{T} \eta_t |c_t|^\top x_t + \frac{D}{\eta_{T+1}}. \tag{C.1}
$$

Bouding the first term, we have

$$
\begin{aligned}
\sum_{t=1}^{T} \eta_t |c_t|^\top x_t &= \sum_{t=1}^{T} \sqrt{D} \cdot \frac{|c_t|^\top x_t}{\sqrt{D + \sum_{j<t} |c_t|^\top x_t}} \\
&\leq 2\sqrt{D} \cdot \sqrt{\sum_{t=1}^{T} |c_t|^\top x_t}, \tag{C.2}
\end{aligned}
$$

using Proposition A.6 with $a_t = |c_t|^\top x_t$ and $u = D \leq 1$. Next,

$$
\begin{aligned}
\frac{D}{\eta_{T+1}} &= \sqrt{D} \cdot \sqrt{D + \sum_{t=1}^{T} |c_t|^\top x_t} \\
&\leq D + \sqrt{D} \cdot \sqrt{\sum_{t=1}^{T} |c_t|^\top x_t}. \tag{C.3}
\end{aligned}
$$

Plugging Eq. (C.2) and Eq. (C.3) into Eq. (C.1) gives

$$
\sum_{t=1}^{T} \left( c_t^\top x_t - c_t^\top z \right) \leq 3\sqrt{D} \sqrt{\sum_{t=1}^{T} |c_t|^\top x_t} + D.
$$

$\qquad\square$

## D  Additional Proofs from Section 3

### D.1  Proof of Lemma 3.2

Following [12], we define the function

$$
\Psi(s) := e^{s-1} \sum_{t=1}^{T} G_t(x_t(s)) + \sum_{t=1}^{T} \ell_t(x_t(s))
$$

for $s \in [0, 1]$ where $G_t$ is the multilinear extension of $g_t$.

We will need the following two lemmas.

**Lemma D.1** (Feldman [12, Lemma 3.2]).

$$\frac{\mathrm{d}\Psi(s)}{\mathrm{d}s} = e^{s-1}\sum_{t=1}^{T} G_t(x_t(s)) + \sum_{t=1}^{T}(e^{s-1}\nabla G_t(x_t(s)) + \ell_t)^\top y_t(s).$$

**Lemma D.2.** *For $s \in [0,1)$,*

$$e^{s-1}\sum_{t=1}^{T} G_t(x_t(s)) + \sum_{t=1}^{T}(e^{s-1}\nabla G_t(x_t(s)) + \ell_t)^\top y_t(s) \geq e^{s-1}\sum_{t=1}^{T} g_t(X^*) + \sum_{t=1}^{T}\ell_t(X^*) - r_s.$$

*Proof.* By the definition of the regret $r_s$,

$$\sum_{t=1}^{T}(e^{s-1}\nabla G_t(x_t(s)) + \ell_t)^\top \mathbf{1}_{X^*} - \sum_{t=1}^{T}(e^{s-1}\nabla G_t(x_t(s)) + \ell_t)^\top y_t(s) \leq r_s.$$

Using the properties of the multilinear extension,

$$\sum_{t=1}^{T}[g_t(S^*) - G_t(x_t(s))] \leq \sum_{t=1}^{T}[G_t(x_t(s) \vee \mathbf{1}_{S^*}) - G_t(x_t(s))]$$
$$\text{(since } g_t(X^*) \leq G_t(x_t(s) \vee \mathbf{1}_{X^*}) \text{ by monotonicity)}$$
$$\leq \sum_{t=1}^{T}\nabla G_t(x_t(s))^\top(x_t(s) \vee \mathbf{1}_{S^*} - x_t(s))$$
$$\text{(since } G_t \text{ is concave along nonnegative directions)}$$
$$\leq \sum_{t=1}^{T}\nabla G_t(x_t(s))^\top \mathbf{1}_{S^*}.$$
$$\text{(since } x_t(s) \vee \mathbf{1}_{S^*} - x_t(s) \leq \mathbf{1}_{S^*} \text{ and } \nabla G_t(x_t(s)) \geq 0)$$

Combining these two inequalities,

$$e^{s-1}\sum_{t=1}^{T} G_t(x_t(s)) + \sum_{t=1}^{T}(e^{s-1}\nabla G_t(x_t(s)) + \ell_t)^\top y_t(s)$$
$$\geq e^{s-1}\sum_{t=1}^{T} G_t(x_t(s)) + \sum_{t=1}^{T}(e^{s-1}\nabla G_t(x_t(s)) + \ell_t)^\top \mathbf{1}_{S^*} - r_s$$
$$= e^{s-1}\sum_{t=1}^{T}[G_t(x_t(s)) - \nabla G_t(x_t(s))^\top \mathbf{1}_{S^*}] + \sum_{t=1}^{T}\ell_t(S^*) - r_s$$
$$\geq e^{s-1}\sum_{t=1}^{T} g_t(S^*) + \sum_{t=1}^{T}\ell_t(S^*) - r_s.$$

$\square$

*Proof of Lemma 3.2.* By Lemma D.1 and Lemma D.2, we have

$$\frac{\mathrm{d}\Psi(s)}{\mathrm{d}s} \geq e^{s-1}\sum_{t=1}^{T} g_t(S^*) + \sum_{t=1}^{T}\ell_t(X^*) - r_s.$$

for $s \in [0,1]$. Integrating this from 0 to 1,

$$\Psi(1) - \Psi(0) \geq (1 - 1/e)\sum_{t=1}^{T} g_t(S^*) + \sum_{t=1}^{T}\ell_t(S^*) - R,$$

where $R := \int_0^1 r_s \mathrm{d}s$. Since $\Psi(1) - \Psi(0) = \sum_{t=1}^T G_t(x_t) + \sum_{t=1}^T \ell_t(x_t)$, we obtain

$$(1 - 1/e) \sum_{t=1}^T g_t(S^*) + \sum_{t=1}^T \ell_t(S^*) - \sum_{t=1}^T G_t(x_t) - \sum_{t=1}^T \ell_t(x_t) \leq R.$$

Now the desired approximation ratio follows from

$$(1 - 1/e) \sum_{t=1}^T g_t(S^*) + \sum_{t=1}^T \ell_t(S^*)$$

$$= (1 - 1/e) \sum_{t=1}^T f_t(S^*) + 1/e \sum_{t=1}^T \ell_t(S^*)$$

$$\geq (1 - 1/e) \sum_{t=1}^T f_t(S^*) + (1 - c)/e \sum_{t=1}^T f_t(S^*)$$

$$\geq (1 - c/e) \sum_{t=1}^T f_t(S^*).$$

Finally, we apply an oblivious rounding to $x_t$, we obtain

$$(1 - c/e) \sum_{t=1}^T f_t(S^*) - \mathbf{E}\left[\sum_{t=1}^T f_t(S_t)\right] \leq R,$$

as desired. □

### D.2 Proof of Claim 3.4

*Proof.* By Lemma D.1, we have

$$\sum_{t=1}^T (e^{s-1} \nabla G_t(x_t(s)) + \ell_t)^\top y_t(s)$$

$$\leq e^{s-1} \sum_{t=1}^T G_t(x_t(s)) + \sum_{t=1}^T (e^{s-1} \nabla G_t(x_t(s)) + \ell_t)^\top y_t(s)$$

$$= \frac{\mathrm{d}\Psi(s)}{\mathrm{d}s}.$$

Thus,

$$\rho = \int_0^1 \sum_{t=1}^T (e^{s-1} \nabla G_t(x_t(s)) + \ell_t)^\top y_t(s)$$

$$\leq \Psi(1) - \Psi(0)$$

$$\leq \sum_{t=1}^T (G_t(x_t(1)) + \ell_t(x_t(s)))$$

$$= \sum_{t=1}^T F_t(x_t(1))$$

$$\leq T.$$

□

### D.3 Bregman projection onto the matroid base polytope

In this section, we will denote a matroid by $\mathcal{M} = (E, \mathcal{I})$ where $E$ is the groundset and $\mathcal{I} \subseteq 2^E$ are the independent sets. Algorithm 5 is a specialized form of the algorithm from [17]. Recall that the

generalized KL divergence is defined as

$$D_{\text{KL}}(x, y) = \sum_{e \in E} x_e \ln \frac{x_e}{y_e} - x_e + y_e.$$

We will write $\Pi_P^{\text{KL}}(y) := \text{argmin}_{x \in P} D_{\text{KL}}(x, y)$ to be the projection of $y$ onto $P$ under KL divergence.

---

**Algorithm 5** Bregman Projection onto Matroid Base Polytope

---

**Input:** $y \in \mathbb{R}_{>0}^E$, matroid $\mathcal{M} = (E, \mathcal{I})$
**Output:** $x^* \in \text{argmin}_{x \in B(\mathcal{M})} D_{\text{KL}}(x, y)$
  1: Initialize $x^{(0)} \leftarrow \frac{y}{n\|y\|_1}$, $N_1 \leftarrow E$, $t \leftarrow 0$.
  2: **while** $N_t \neq \emptyset$ **do**
  3:    Define $z \in \mathbb{R}^E$ by

$$z_e = \begin{cases} x_e^{(t)} & e \in N_t \\ 0 & e \notin N_t \end{cases}.$$

  4:    $\delta_{t+1} \leftarrow \max\{\delta : x^{(t)} + \delta z \in B_{\mathcal{M}}\}$.
  5:    $x^{(t+1)} \leftarrow x^{(t)} + \delta_{t+1} z$.
  6:    Let $F_{t+1} \subseteq N_t$ be a maximal set such that $x^{(t+1)}(F_1 \cup \ldots \cup F_{t+1}) = \text{rk}(F_1 \cup \ldots \cup F_{t+1})$.
  7:    $N_{t+1} \leftarrow N_t \setminus F_{t+1}$.
  8:    $t \leftarrow t + 1$
  9: **return** $x^{(t)}$

---

**Lemma D.3** (Gupta et al. [17, Theorem 3]). *For all $y \in \mathbb{R}_{>0}^E$, Algorithm 5 outputs $\Pi_{B(\mathcal{M})}^{\text{KL}}(y)$.*

Lemma D.3 is stated in more generality in [17]. To keep this paper as self-contained as possible, we will prove Lemma D.3 in our special case (although the proof itself follows that in [17]). We will require the following lemma which is a consequence of the fact that the greedy algorithm optimizes linear functions over the matroid base polytope.

**Lemma D.4.** *Let $\mathcal{M} = (E, \mathcal{I})$ be a matroid and let $B_{\mathcal{M}}$ be the base polytope. Let $w \in \mathbb{R}^E$. Consider the (unique) disjoint partitioning of $E = \cup_{i=1}^k F_i$ satisfying:*

  1. *$F_1, \ldots, F_k \neq \emptyset$;*
  2. *if $e, e' \in F_i$ then $w_e = w_{e'}$;*
  3. *if $e_i \in F_i, e_j \in F_j$ and $i < j$ then $w_i < w_j$; and*

*Then $x^* \in \text{argmin}_{w \in B(\mathcal{M})} w^\top x$ if and only if*

$$x^*(F_1 \cup \ldots \cup F_i) = \text{rk}(F_1 \cup \ldots \cup F_i)$$

*for every $i \in [k]$.*

*Proof of Lemma D.3.* Let $h(x) := \sum_{e \in E} x_e \ln \frac{x_e}{y_e} - x_e + y_e$. Then $x^{(t)} \in \text{argmin}_{x \in B(\mathcal{M})} h(x) = D_{\text{KL}}(x, y)$ if and only if $\nabla h(x^{(t)})^\top (x - x^{(t)}) \geq 0$ for all $x \in B(\mathcal{M})$. In other words, we require that $x^{(t)} \in \text{argmin}_{x \in B(\mathcal{M})} \nabla h(x^{(t)})^\top x$. In the rest of the proof, we will verify that this inclusion holds for the return point of Algorithm 5.

Suppose that Algorithm 5 terminates after $t$ iterations. Let $F_1, \ldots, F_t$ be the sets constructed in Algorithm 5. By construction $F_1, \ldots, F_t$ form a disjoint partition of $E$. Recall that $\nabla h(x^{(t)})_e = \ln \frac{x_e^{(t)}}{y_e}$. By construction, if $e \in F_i$ then

$$x_e^{(t)} = c y_e (\delta_1 + \ldots + \delta_i)$$

where $c = \frac{1}{n\|y\|_1}$. Hence,

$$\nabla h(x^{(t)})_e = \ln(c) + \ln(\delta_1 + \ldots + \delta_i) \tag{D.1}$$

for $e \in F_i$. Note that the RHS of Eq. (D.1) is strictly increasing in $i$ and by construction, $x^{(t)}(F_1 \cup \ldots \cup F_i) = \text{rk}(F_1 \cup \ldots \cup F_i)$ for all $i \in [t]$. Lemma D.4 then implies that

$$x^{(t)} \in \underset{x \in B(\mathcal{M})}{\text{argmin}} \nabla(h(x^{(t)}))^\top (x - x^{(t)}),$$

which, as asserted above, implies that $x^{(t)} \in \text{argmin}_{x \in B(\mathcal{M})} D_{\text{KL}}(x, y)$. $\qquad \square$

**Theorem D.5.** *There is a polynomial-time algorithm for computing Bregman projection onto a matroid base polytope.*

*Proof.* Line 4 of Algorithm 5 can be implemented in polynomial-time (see e.g. [15, Theorem 2]). Line 6 can be computed by finding the unique maximal minimizer of the submodular function $\text{rk}(\cdot) - x^{(t+1)}(\cdot)$ [22, Theorem 3.1]. The correctness of the algorithm follows from Lemma D.3. $\quad \square$

**Remark D.6.** *It is possible to generalize Theorem D.5 for arbitrary mirror maps and general polymatroids. The details can be found in [17].*

# E  Discrete version of Algorithm 1

In this section, we describe a discrete version of Algorithm 1 and formally prove Theorem 3.1.

---

**Algorithm 6** Discrete time algorithm

---

**Input:** accuracy $\varepsilon > 0$
1: Take the largest $\delta \in (0, \varepsilon/n^2]$ such that $1/\delta$ is a positive integer.
2: **for** $s = 0, \delta, 2\delta, \ldots, 1 - \delta$ **do**
3:      Initialize online dual averaging algorithms $\mathcal{A}_s$ over matroid base polytope $B_{\mathcal{M}}$.
4: **for** $t = 1, 2, \ldots$ **do**
5:      Set $x_t(0) = \mathbf{0}$.
6:      **for** $s = 0, \delta, 2\delta, \ldots, 1 - \delta$ **do**
7:          Set $x_t(s + \delta) = x_t(s) + \delta \cdot y_t(s)$, where $y_t(s) \in P_{\mathcal{M}}$ is the prediction provided by $\mathcal{A}_s$.
8:      Apply swap rounding to $x_t := x_t(1)$ and obtain $S_t$.
9:      Play $S_t$ and observe $f_t$.
10:      Compute the modular function $\ell_t$ for $f_t$ by (3.1) and let $g_t = f_t - \ell_t$.
11:      **for** $s = 0, \delta, 2\delta, \ldots, 1 - \delta$ **do**
12:          Compute an estimator $\nabla_t(s)$ of $\nabla G_t(x_t(s))$ by using $O(n^2 \varepsilon^{-2} \log(\frac{nt}{\delta}))$ samples.
13:          Feedback the reward vector $c_t = -(1 + \delta)^{(s-1)/\delta} \cdot \nabla_t(s) - \ell_t$ to $\mathcal{A}_s$.

---

For the analysis, let us fix $T > 0$. Let $M_t = \max_{i \in V} f_t(i)$. Using a standard Chernoff bound argument (see Feldman [12, Lemma A.3]) we see that, for all $t$,

$$E_t := n \cdot \max_s \|\nabla_t(s) - \nabla G_t(x_t(s))\|_\infty \leq \varepsilon M_t \tag{E.1}$$

holds with probability at least $1 - 1/nt^2$. Following [12], we define $\Psi(s) = \sum_{t=1}^{T}[(1 + \delta)^{(s-1)/\delta} G(x_t(s)) + \ell_t^\top x_t(s)]$. Let us fix $S^*$ to be an arbitrary optimal solution. We will also write $M = \sum_{t=1}^{T} M_t$ and $E = \sum_{t=1}^{T} E_t$.

The first lemma is adapted from the proof of Lemma A.5 in [12] but where we carry around the error terms in Eq. (E.1).

**Lemma E.1** (Feldman [12, Lemma A.5])**.**

$$\frac{\Psi(s + \delta) - \Psi(s)}{\delta}$$

$$\geq \sum_{t=1}^{T}(1 + \delta)^{(s-1)/\delta} G(x_t(s)) + y_t(s)^\top \left[ (1 + \delta)^{(s-1)/\delta} \nabla_t(s) + \ell_t \right] - \varepsilon M - E.$$

**Lemma E.2.** *For each $s$,*

$$\sum_{t=1}^{T} y_t(s)^\top \left[(1+\delta)^{(s-1)/\delta} \nabla_t(s) + \ell_t\right]$$

$$\geq \sum_{t=1}^{T} (1+\delta)^{(s-1)/\delta} [g_t(S^*) - G_t(x_t(s))] + \ell_t(S^*) - r_s - E,$$

*where $r_s$ is the regret of $\mathcal{A}_s$.*

*Proof.* By the definition of the regret $r_s$, we have

$$\sum_{t=1}^{T} y_t(s)^\top \left[(1+\delta)^{(s-1)/\delta} \cdot \nabla_t(s) + \ell_t\right]$$

$$\geq \sum_{t=1}^{T} \mathbf{1}_{S^*}^\top \left[(1+\delta)^{(s-1)/\delta} \cdot \nabla_t(s) + \ell_t\right] - r_s$$

$$= \sum_{t=1}^{T} \left[(1+\delta)^{(s-1)/\delta} \cdot \mathbf{1}_{S^*}^\top \nabla_t(s) + \ell_t(S^*)\right] - r_s$$

$$\geq \sum_{t=1}^{T} \left[(1+\delta)^{(s-1)/\delta} \cdot \mathbf{1}_{S^*}^\top \nabla G_t(x_t(s)) - E_t + \ell_t(S^*)\right] - r_s$$

$$\geq \sum_{t=1}^{T} \left[(1+\delta)^{(s-1)/\delta} \cdot [g_t(S^*) - G_t(x_t(s))] + \ell_t(S^*)\right] - r_s - E,$$

where we used the similar analysis as in the continuous case in the last inequality. $\square$

Combining these two lemmas, we have

$$\frac{\Psi(s+\delta) - \Psi(s)}{\delta} \geq \sum_{t=1}^{T} (1+\delta)^{(s-1)/\delta} g_t(S^*) + \ell_t(S^*) - r_s - \varepsilon M - 2E.$$

for each $s$. Summing up this for $s$, we obtain

$$\Psi(1) - \Psi(0) \geq \sum_{t=1}^{T} [C(\delta) g_t(S^*) + \ell_t(S^*)] - \delta \sum_s r_s - \varepsilon M - 2E,$$

where $C(\delta) \coloneqq \sum_s \delta(1+\delta)^{(s-1)/\delta} \geq 1 - 1/e - \varepsilon/n$, provided that $\delta \leq \varepsilon/n^2$ (see [12, Proof of Lemma A.8]). Since $g_t(S^*) \leq n \max_{i \in S^*} g_t(i) \leq nM_t$, we have

$$\Psi(1) - \Psi(0) \geq \sum_{t=1}^{T} [(1-1/e) g_t(S^*) + \ell_t(S^*)] - \delta \sum_s r_s - 2\varepsilon M - 2E.$$

Thus, following the same argument as in the continuous case, we obtain

$$(1 - c/e) \sum_{t=1}^{T} f_t(S^*) - \sum_{t=1}^{T} F_t(x_t) - 2\varepsilon M - 2E \leq \delta \sum_s r_s. \tag{E.2}$$

Next, we will need a small claim bounding $\mathbf{E}[E]$.

**Claim E.3.** $\mathbf{E}[E] \leq \varepsilon M + O(1)$.

*Proof.* For any $t$, $E_t \leq \varepsilon M_t$ with probability $1 - 1/nt^2$. With the remaining $1/nt^2$ probability, we have a trivial upper bound of $E_t \leq n$. Hence, $\mathbf{E}[E_t] \leq \varepsilon M_t + 1/t^2$. Summing up over $t$ gives $\mathbf{E}[E_t] \leq \varepsilon M + O(1)$. $\square$

Hence, taking expectations in Eq. (E.2) and using the property of swap rounding, we have

$$(1 - c/e) \sum_{t=1}^{T} f_t(S^*) - \sum_{t=1}^{T} \mathbf{E}[f_t(S_t)] - 4\varepsilon M - O(1) \leq \delta \sum_s r_s.$$

As $M \leq \sum_{t=1}^{T} f_t(S^*)$, we thus have

$$(1 - c/e - 4\varepsilon) \sum_{t=1}^{T} f_t(S^*) - \sum_{t=1}^{T} \mathbf{E}[f_t(S_t)] - O(1) \leq \delta \sum_s r_s =: R.$$

It remains to bound $R$. To that end, define

$$\rho_s := \sum_{t=1}^{T} \left[ (1 + \delta)^{(s-1)/\delta} \nabla_t(s) + \ell_t \right]^{\top} y_t(s),$$

which is the reward received by algorithm $\mathcal{A}_s$. As in the continuous case, suppose each $\mathcal{A}_s$ is an instance of an online dual averaging algorithm (Algorithm 4) with initial point $y_1(s) = \Pi_{\Phi}^{\mathrm{KL}} \left( \frac{k}{n} \mathbf{1} \right)$. Here $\Phi$ is the negative entropy mirror map. Fact A.4 and Fact A.5 imply that $\sup_{u \in \mathcal{X}} D_{\mathrm{KL}} (u, x_1) \leq k \ln(n/k)$. Hence, using Corollary B.3 (applied with $c_t = -e^{s-1} \nabla G_t(y_t(s)) - \ell_t \in \mathbb{R}_{\leq 0}^n$ and $D = k \ln(n/k)$), we have

$$r_s \leq 3\sqrt{k \ln(n/k)} \sqrt{\rho_s} + k \ln(n/k). \tag{E.3}$$

The following lemma bounds the regret.

**Lemma E.4.** *Using an OLO algorithm that guarantees Eq.* (E.3)*, we have*
$$R \leq O(\sqrt{k \ln(n/k)} \sqrt{T}).$$

Before we prove Lemma E.4, we will need a claim to bound $\delta \sum_s \rho_s$.

**Claim E.5.** $\delta \sum_s \mathbf{E}[\rho_s] \leq O(T)$.

The proof of Claim E.5 can be found below.

*Proof of Lemma E.4.* If $T \leq k \ln(n/k)$ then we trivially bound $r_s \leq T \leq \sqrt{k \ln(n/k)} \sqrt{T}$. Since $R = \delta \sum_s r_s$, we have $R \leq \sqrt{k \ln(n/k)} \sqrt{T}$ if $T \leq k \ln(n/k)$. Henceforth, we assume $T \geq k \ln(n/k)$. Summing over $s = 0, \delta, \ldots, 1 - \delta$, we have

$$R = \delta \sum_s r_s \leq 3\sqrt{k \ln(n/k)} \sum_s \delta \sqrt{\rho_s} + k \ln(n/k)$$

$$\leq 3\sqrt{k \ln(n/k)} \sqrt{\sum_s \delta \rho_s} + k \ln(n/k) \quad \text{(Jensen's Ienquality)}.$$

Taking expectations and applying Jensen's Inequality, we get that

$$\mathbf{E}[R] \leq 3\sqrt{k \ln n(/k)} \mathbf{E} \left[ \sqrt{\sum_s \delta \rho_s} \right] + k \ln(n/k)$$

$$\leq 3\sqrt{k \ln(n/k)} \sqrt{\sum_s \delta \mathbf{E}[\rho_s]} + k \ln(n/k)$$

$$\leq O(\sqrt{k \ln(n/k)} \sqrt{T}),$$

where in the last inequality we used Claim E.5 and our assumption that $T \geq k \ln(n/k)$. □

*Proof of Claim E.5.* By Lemma E.1, we have $\delta \rho_s \leq \Psi(s + \delta) - \Psi(s) + \delta \varepsilon M + \delta E$. Summing over all $s = 0, \delta, \ldots, 1 - \delta$ gives

$$\delta \sum_s \rho_s \leq \Psi(1) - \Psi(0) + \varepsilon M + E$$

$$\leq \sum_{t=1}^{T} [G(x_t(1)) + \ell_t^{\top} x_t(1)] + \varepsilon M + E$$

$$\leq T + \varepsilon M + E.$$

Note that $M \leq T$. Taking expectations and applying Claim E.3 to bound $\mathbf{E}[E]$ gives $\delta \sum_s \mathbf{E}[\rho_s] \leq (1 + 2\varepsilon)T + O(1) \leq O(T)$. □

# F  Additional Proofs from Section 4

## F.1  Proof of Theorem 4.1

If $T \leq n$ then we have a trivial regret bound of $T \leq \sqrt{nT}$. Henceforth, we assume that $T \geq n$. Recall that $r_i(T) = \max\left\{\sum_{t=1}^T p_{t,i}^- \Delta_{t,i}^+, \sum_{t=1}^T p_{t,i}^+ \Delta_{t,i}^-\right\} - \frac{1}{2}\sum_{t=1}^T \left(p_{t,i}^+ \Delta_{t,i}^+ + p_{t,i}^- \Delta_{t,i}^-\right)$. Let $g_i = \max\left\{\sum_{t=1}^T p_{t,i}^- |\Delta_{t,i}^+|, \sum_{t=1}^T p_{t,i}^+ |\Delta_{t,i}^-|\right\}$. By Lemma 4.8 and Lemma 4.4,

$$r_i(T) \leq O\left(\sqrt{g_i} + \sqrt{\sum_{t \in C_i^+ \cap [T]} \alpha_{t,i}} + \sqrt{\sum_{t \in C_i^- \cap [T]} \beta_{t,i}} + 1\right) \tag{F.1}$$

where $C_i^+, C_i^-, \alpha_{t,i}, \beta_{t,i}$ are as defined in Lemma 4.4. The following two lemmas bounds each of the terms in Eq. (F.1). We relegate the proofs to Appendix F.4.

**Lemma F.1.** *The following two bounds hold:*

    *1.* $\mathbf{E}[\sum_{i=1}^n \sqrt{\sum_{t \in C_i^+ \cap [T]} \alpha_{t,i}}] \leq O(\sqrt{nT})$; *and*

    *2.* $\mathbf{E}[\sum_{i=1}^n \sqrt{\sum_{t \in C_i^- \cap [T]} \beta_{t,i}}] \leq O(\sqrt{nT})$.

**Lemma F.2.** $\sum_{i=1}^n \mathbf{E}\sqrt{g_i} \leq O(\sqrt{nT})$.

*Proof of Theorem 4.1.* By Lemma 4.3, it suffices to bound $\sum_{i=1}^n \mathbf{E}[r_i(T)]$. Using Eq. (F.1), Lemma F.1, and Lemma F.2, we have $\sum_{i=1}^T \mathbf{E}[r_i(T)] \leq O(\sqrt{nT}) + O(n) \leq O(\sqrt{nT})$, where the last inequality is because $n \leq \sqrt{nT}$. □

## F.2  Details of halfspace oracles for the Blackwell instances in Section 4

We describe how to construct an efficient halfspace oracle for the Blackwell instances corresponding to USM balance subproblems in Section 4 via strong duality of LP. We use the same notation from Section 4. Let us assume that a halfspace $H$ is given by a linear inequality $a^\top z \leq \beta$ for some $a \in \mathbb{R}^2$ and $\beta \in \mathbb{R}$. Since $H$ contains $\mathbb{R}_{\leq 0}^2$, one can assume $\beta = 0$ without loss of generality. Then, $p \in \mathcal{X}$ is a valid output of an half-space oracle if $\max_{\Delta \in \mathcal{Y}} a^\top u(p, \Delta) \leq 0$. Therefore, to find such $p$, it suffices to solve the min-max linear programming

$$\min_{p \in \mathcal{X}} \max_{\Delta \in \mathcal{Y}} a^\top u(p, \Delta).$$

Now, replacing the inner maximization with the dual problem, we have an equivalent LP

$$\begin{aligned}
\min_{p,z} \quad & z_1 + z_2 - z_3 - z_4 \\
\text{subject to} \quad & p \in \mathcal{X} \\
& -z_0 + z_1 - z_3 = a^+ \cdot p^- - p^+ \\
& -z_0 + z_2 - z_4 = a^- \cdot p^+ - p^- \\
& z_0, z_1, z_2, z_3, z_4 \geq 0,
\end{aligned} \tag{F.2}$$

where we used $a^\top u(p, \Delta) = (a^+ \cdot p^- - p^+, a^- \cdot p^+ - p^-)^\top \Delta$. Since it is a constant dimensional problem, one can solve it in $O(1)$ time.

### F.3 Proof of Claim 4.9 and Lemma 4.10

*Proof of Claim 4.9.* If $x \in B_2(1) \cap \mathbb{R}_{\geq 0}$ then

$$
\begin{aligned}
D_{\mathrm{KL}}(x, x_1) &= x^+ \ln(\sqrt{2}x^+) + x^- \ln(\sqrt{2}x^-) - \|x\|_1 + \sqrt{2} \\
&\leq x^+ \ln(x^+) + x^- \ln(x^-) + \sqrt{2}\ln(\sqrt{2}) + \sqrt{2} \\
&\leq \sqrt{2}\ln(\sqrt{2}) + \sqrt{2} \leq 2.
\end{aligned}
$$

The second last inequality is because $x^+, x^- \in [0, 1]$ so $\ln(x^+), \ln(x^-) \leq 0$. $\qquad\square$

*Proof of Lemma 4.10.* First, we use the trivial upper bound $|c_t|^\top x_t \leq |c_t^+| + |c_t^-|$. We now bound $|c_t^+|$ and $|c_t^-|$ separately. Suppose first that $t \in C^+$. In this case $|c_t^+| = c_t^+$. Using the bound $-\Delta_t^- \leq \Delta_t^+$, we have

$$
c_t^+ \leq \frac{1}{2}(p_t^+ \cdot \Delta_t^+ + p_t^- \cdot \Delta_t^-) + p_t^+ \cdot \Delta_t^+.
$$

On the other hand, if $t \notin C^+$ then $|c_t^+| = -c_t^+$. Using that $-\Delta_t^+ \leq \Delta_t^-$ and $-\Delta_t^- \leq \Delta_t^+$, we have

$$
-c_t^+ \leq p_t^+ \Delta_t^- + \frac{1}{2}(p_t^+ \Delta_t^- + p_t^- \Delta_t^+) \leq \frac{3}{2}p_t^+ |\Delta_t^-| + \frac{1}{2}p_t^- |\Delta_t^+|.
$$

Hence,

$$
\begin{aligned}
|c_t^+| &\leq \left(\frac{3}{2}p_t^+ \cdot \Delta_t^+ + \frac{1}{2}p_t^- \cdot \Delta_t^-\right)\mathbf{1}[t \in C^+] + \left(\frac{3}{2}p_t^+ |\Delta_t^-| + \frac{1}{2}p_t^- |\Delta_t^+|\right)\mathbf{1}[t \notin C^+] \\
&\leq \left(\frac{3}{2}p_t^+ \cdot \Delta_t^+ + \frac{1}{2}p_t^- \cdot \Delta_t^-\right)\mathbf{1}[t \in C^+] + \left(\frac{3}{2}p_t^+ |\Delta_t^-| + \frac{1}{2}p_t^- |\Delta_t^+|\right).
\end{aligned}
$$

With nearly identical reasoning, we have

$$
|c_t^-| \leq \left(\frac{1}{2}p_t^+ \cdot \Delta_t^+ + \frac{3}{2}p_t^- \cdot \Delta_t^-\right)\mathbf{1}[t \in C^-] + \left(\frac{1}{2}p_t^+ |\Delta_t^-| + \frac{3}{2}p_t^- |\Delta_t^+|\right).
$$

We conclude that

$$
\begin{aligned}
|c_t^+| + |c_t^-| \leq{}& \left(\frac{3}{2}p_t^+ \cdot \Delta_t^+ + \frac{1}{2}p_t^- \cdot \Delta_t^-\right)\mathbf{1}[t \in C^+] + \left(\frac{1}{2}p_t^+ \cdot \Delta_t^+ + \frac{3}{2}p_t^- \cdot \Delta_t^-\right)\mathbf{1}[t \in C^-] \\
&+ 2(p_t^+ |\Delta_t^-| + p_t^- |\Delta_t^+|).
\end{aligned}
$$

Summing up the right hand side of the bound gives the claim. $\qquad\square$

### F.4 Proof of Lemma F.1 and Lemma F.2

In this section, we let $\mathcal{F}_{t,i}$ denote the $\sigma$-algebra containing all randomness up to the $i$th iteration at time $t$.[7]

*Proof of Lemma F.1.* We prove only the first inequality. The second inequality is nearly identical. Now,

$$
\begin{aligned}
\mathbf{E}\left[\sum_{i=1}^n \sqrt{\sum_{t \in C_i^+ \cap [T]} \alpha_{t,i}}\right] &\leq \sqrt{n}\sqrt{\mathbf{E}\left[\sum_{i=1}^n \sum_{t \in C_i^+ \cap [T]} \alpha_{t,i}\right]} \quad \text{(Cauchy-Schwarz)} \\
&= \sqrt{n}\sqrt{\mathbf{E}\left[\sum_{i=1}^n \sum_{t \in C_i^+ \cap [T]} \left(\frac{3}{2}p_t^+ \Delta_t^+ + \frac{1}{2}p_t^- \Delta_t^-\right)\right]}
\end{aligned}
$$

As asserted in Lemma 4.4, the event $t \in C_i^+$ depends only on $p_{t,i}, \Delta_{t,i}$ both of which are $\mathcal{F}_{t,i-1}$-measurable. Hence, applying Claim F.3 gives $\mathbf{E}\left[\sum_{i=1}^n \sqrt{\sum_{t \in C_i^+ \cap [T]} \alpha_{t,i}}\right] \leq 2\sqrt{nT}$. $\qquad\square$

**Claim F.3.** *Let $S_t \subseteq [n]$ be a random set such that the event $\{i \in S_t\}$ can be determined by knowing $\Delta_{t,i}$ and $p_{t,i}$. Then $\mathbf{E}[\sum_{i \in S_t} p_{t,i}^+ \Delta_{t,i}^+] \leq 1$ and $\mathbf{E}[\sum_{i \in S_t} p_{t,i}^- \Delta_{t,i}^-] \leq 1$.*

*Proof.* We prove only the first inequality as the second inequality is similar. Recall that $\mathcal{F}_{t,i}$ is the $\sigma$-algebra generated by all randomness up iteration $i$ of the algorithm at time $t$. Then $\Delta_{t,i}$ and $p_{t,i}$ are $\mathcal{F}_{t,i-1}$-measurable so $\{i \in S_t\}$ is $\mathcal{F}_{t,i-1}$-measurable. Thus

$$
\mathbf{E}[\sum_{i \in S_t} p_{t,i}^+ \Delta_{t,i}^+] = \mathbf{E}\big[\sum_{i=1}^n p_{t,i}^+ \Delta_{t,i}^+ \mathbf{1}[i \in S_t]\big]
$$

$$
= \mathbf{E}\big[\sum_{i=1}^n \mathbf{E}[f_t(X_{t,i}) - f_t(X_{t,i-1}) \mid \mathcal{F}_{t,i-1}]\mathbf{1}[i \in S_t]\big]
$$

$$
= \mathbf{E}\big[\mathbf{E}[\sum_{i=1}^n (f_t(X_{t,i}) - f_t(X_{t,i-1}))\mathbf{1}[i \in S_t]\big] \quad (\mathbf{1}[i \in S_t] \text{ is } \mathcal{F}_{t,i-1}\text{-measurable})
$$

$$
= \mathbf{E}\big[\sum_{i \in S_t} f_t(X_{t,i}) - f_t(X_{t,i-1})\big]
$$

$$
\leq 1,
$$

where the last inequality is by Claim A.3. $\qquad\square$

We now turn to the proof of Lemma F.2. Define $N_i^+ := \{t \in [T] : \Delta_{t,i}^+ < 0\}$ and $N_i^- := \{t \in [T] : \Delta_{t,i}^- < 0\}$. Recall that

$$
g_i = \max\left\{\sum_{t=1}^T p_{t,i}^- |\Delta_{t,i}^+|, \sum_{t=1}^T p_{t,i}^+ |\Delta_{t,i}^-|\right\}.
$$

The following simple claim will prove to be useful.

**Claim F.4.**

$$
\max\left\{\sum_{t=1}^T p_{t,i}^- \Delta_{t,i}^+, \sum_{t=1}^T p_{t,i}^+ \Delta_{t,i}^-\right\} \geq g_i - \sum_{t \in N_i^+} 2p_{t,i}^- \cdot \Delta_{t,i}^- - \sum_{t \in N_i^-} 2p_{t,i}^+ \cdot \Delta_{t,i}^+. \tag{F.3}
$$

The proof of Claim F.4 is straightforward manipulations and can be found below.

*Proof of Lemma F.2.* Recalling the definition of $r_i(T)$ (from Eq. (4.1)) and applying Lemma 4.8 we have

$$
\max\left\{\sum_{t=1}^T p_{t,i}^- \Delta_{t,i}^+, \sum_{t=1}^T p_{t,i}^+ \Delta_{t,i}^-\right\} - \frac{1}{2}\sum_{t=1}^T (p_{t,i}^+ \Delta_{t,i}^+ - p_{t,i}^- \Delta_{t,i}^-) \leq \mathrm{Reg}_{\mathcal{A}_i}(T). \tag{F.4}
$$

Using Claim F.4 and Claim F.3 to lower bound the left-hand side of Eq. (F.4) gives

$$
\sum_{i=1}^n \mathbf{E}[g_i] - C \cdot T \leq \sum_{i=1}^n \mathbf{E}[\mathrm{Reg}_{\mathcal{A}_i}(T)],
$$

for some constant $C > 0$. Hence, using Lemma 4.4 to bound $\mathrm{Reg}_{\mathcal{A}_i}(T)$ and applying Claim F.3 and Lemma F.1, we have, for some (different) constant $C > 0$,

$$
\sum_{i=1}^n \mathbf{E}[g_i] - C \cdot T \leq C \sum_{i=1}^n \sqrt{\mathbf{E}[g_i]} \leq C\sqrt{n}\sqrt{\sum_{i=1}^n \mathbf{E}[g_i]},
$$

where the second inequality is by Jensen's Inequality and the last inequality is by Cauchy-Schwarz. Let $G = \sqrt{\sum_{i=1}^n \mathbf{E}[g_i]}$. The bound becomes $G^2 - CT \leq C\sqrt{n}G$. By Claim A.2,

$$
G \leq \frac{C\sqrt{T} + \sqrt{C^2 n}}{2} \leq O(\sqrt{T}),
$$

where the last inequality is because $n \leq T$. Finally, we have

$$\sum_{i=1}^{n} \mathbf{E}[\sqrt{g_i}] \leq \sqrt{n} \sqrt{\sum_{i=1}^{n} \mathbf{E}[g_i]} = \sqrt{n}G \leq O(\sqrt{nT}),$$

which completes the proof of the lemma. $\qquad\square$

*Proof of Claim F.4.* Note that

$$\sum_{t=1}^{T} p_{t,i}^{-} \cdot \Delta_{t,i}^{+} - \sum_{t \in N_i^+} 2p_{t,i}^{-} \cdot \Delta_{t,i}^{+} = \sum_{t=1}^{T} p_{t,i}^{-} \cdot |\Delta_{t,i}^{+}|.$$

Hence,

$$\sum_{t=1}^{T} p_{t,i}^{-} \cdot \Delta_{t,i}^{+} = \sum_{t=1}^{T} p_{t,i}^{-} \cdot |\Delta_{t,i}^{+}| + \sum_{t \in N_i^+} 2p_{t,i}^{-} \cdot \Delta_{t,i}^{+}$$

$$\geq \sum_{t=1}^{T} p_{t,i}^{-} \cdot |\Delta_{t,i}^{+}| - 2 \sum_{t \in N_i^+} p_{t,i}^{-} \cdot \Delta_{t,i}^{-}$$

where in the last inequality, we used the fact that $\Delta_{t,i}^{+} + \Delta_{t,i}^{-} \geq 0$, which implies that $\Delta_{t,i}^{+} \geq -\Delta_{t,i}^{-}$. Similarly, we have

$$\sum_{t=1}^{T} p_{t,i}^{+} \cdot \Delta_{t,i}^{-} = \sum_{t=1}^{T} p_{t,i}^{+} \cdot |\Delta_{t,i}^{-}| + \sum_{t \in N_i^+} 2p_{t,i}^{+} \cdot \Delta_{t,i}^{-}$$

$$\geq \sum_{t=1}^{T} p_{t,i}^{+} \cdot |\Delta_{t,i}^{-}| - \sum_{t \in N_i^-} 2p_{t,i}^{+} \cdot \Delta_{t,i}^{+}.$$

Hence,

$$\max \left\{ \sum_{t=1}^{T} p_{t,i}^{-} \Delta_{t,i}^{+}, \sum_{t=1}^{T} p_{t,i}^{+} \Delta_{t,i}^{-} \right\} \geq$$

$$\max \left\{ \sum_{t=1}^{T} p_{t,i}^{-} \cdot |\Delta_{t,i}^{+}| - \sum_{t \in N_i^+} 2p_{t,i}^{-} \cdot \Delta_{t,i}^{-}, \sum_{t=1}^{T} p_{t,i}^{+} \cdot |\Delta_{t,i}^{-}| - \sum_{t \in N_i^-} 2p_{t,i}^{+} \cdot \Delta_{t,i}^{+} \right\}.$$

Finally, to get the desired inequality, we use the simple fact that $\max\{\alpha_1 - \beta_1, \alpha_2 - \beta_2\} \geq \max\{\alpha_1, \alpha_2\} - \beta_1 - \beta_2$ whenever $\beta_1, \beta_2 \geq 0$. $\qquad\square$

## Footnotes

[7]Without loss of generality, we assume $f_1, f_2, \dots$ are deterministic (but unknown to the algorithm). If $f_1, f_2, \dots$ are random then we can condition on $f_1, \dots, f_t$ for the argument.