[Reviews · NeurIPS 2020]

Review 1

Summary and Contributions: The paper studies two online submodular maximization problems: monotone with matroid constraint and non-monotone without constraint. In both setting, the paper gives an improved regret bound with a polynomial improvement in either the matroid rank (in the matroid constraint case) or the number of elements (in the unconstrained case). These problems are solved by reducing to multiple online linear optimization problems in parallel and the regret is related to the regret in these problems. The paper performs an improved analysis compared with previous work and bounds the total regret across all coordinates at the same time as opposed to using the per coordinate bound multiplied with the number of coordinates.

Strengths: The paper gives significant improvement in the regret bounds in many different settings. The techniques involved are quite interesting and while building on machinery of general linear online optimization, the bounds are specific to the submodular problems. In previous works, the contribution of each coordinate is simply bounded by the maximum function value and the bound suffers from a factor of d, the dimensions. In this work, especially in the unconstrained setting, heavy lifting is required to add together the contributions from all coordinates and bound them together by the function value.

Weaknesses: No empirical evaluation to see if these improvements hold experimentally compared with previous algorithms.

Correctness: The claims seem believable.

Clarity: The paper is reasonably written.

Relation to Prior Work: The prior works are discussed.

Reproducibility: Yes

Additional Feedback: missing index t on line 193 ---------after rebuttal------------ I remain positive about this paper. The authors addressed all questions raised by the reviewers. I do not see any problem with the results; this is a technically strong paper with nice improvements in the regret bounds in several settings.


Review 2

Summary and Contributions: I remain positive about the paper. === This paper studies online submodular maximization in two settings: 1) monotone submodular function and matroid constraint 2) non-monotone submodular function and unconstrained.

Strengths: In both settings, the proposed algorithms improve the best known ones.

Weaknesses: 1. What is the adversary strength (adversary model) of this paper? Oblivious, adaptive online, or adaptive offline? [16] uses FPL, which only works for an oblivious adversary. I was wondering which adversary model is assumed in this work. This is important since one can make fair comparisons only if the adversary model matches. 2. I was confused by line 133. KL-divergence is defined for probability vectors but here x and y are in R^n. And what if yi = 0 for some i? The authors may want to clarify it here. 3. Is the first-order regret bound just a data-dependent regret bound? For example, is the equation in line 136 a first-order regret bound? It looks similar to guarantees in parameter-free online learning. Are they really related?

Correctness: Yes

Clarity: Yes

Relation to Prior Work: Yes

Reproducibility: Yes

Additional Feedback:


Review 3

Summary and Contributions: This paper studies the online submodular maximization problem in two different settings: i) monotone submodular maximization subject to matroid constraint, ii) unconstrained non-monotone submodular maximization. For the first setting, the authors propose a variant of the online continuous greedy algorithm and obtain O(\sqrt{T}) \alpha-regret bounds where \alpha = 1- c/e is parametrized by the curvature c \in [0,1] of the submodular set function. Therefore, if the utility functions are modular (i.e., c=0), the approximation ratio in the derived regret bounds is 1 and more generally, curvature-independent (1-1/e)-regret bounds could be obtained. For the second setting, the authors propose a variant of online double greedy algorithm through using Blackwell approachability and obtain improved O(\sqrt{T}) (1/2)-regret bounds. --------------------------- Update: I read the authors' response, thanks for your detailed answers. I was convinced by the responses to all the questions except the one about connections to the algorithms of [9] and [10]. I noticed that the regret bounds derived in this work are an improvement (due to using first-order regret bounds), but my question was about the differences of the algorithms themselves. In other words, I was wondering whether the algorithms are the same except the use of the modular functions $\ell_t$ in this work.

Strengths: This work is significant in several aspects. Firstly, the \alpha in the \alpha-regret bounds obtained for the monotone setting is parametrized by the total curvature of the submodular set function c (\alpha = 1-c/e), while in all the previous results, \alpha = 1-1/e holds. Therefore, if the curvature c of the submodular function is strictly smaller than 1, the approximation ratio is better than the previous results. Secondly, although online dual averaging and Blackwell approachability are well-known techniques in the online learning community, exploiting them in the context of submodular maximization has not been done before and is significant. In particular, the use of first-order regret bounds (as opposed to the more common zeroth order bounds) for the online dual averaging and Blackwell approachability subroutines is novel.

Weaknesses: This paper has a few weaknesses which are as follows: - Although the authors have cited [9] and [10] as previous literature on online submodular maximization, they have not compared their obtained regret bounds with the bounds of [9] and [10]. These two papers consider the online continuous submodular maximization problem (as opposed to submodular set function maximization in this work) and may seem different in the first look, however, since the multilinear extension of submodular set functions are continuous submodular, combination of the continuous algorithms in [9] and [10] and a lossless rounding technique such as pipage rounding could be used for discrete problems. In particular, the algorithms of [9] and [10] are almost identical to Algorithm 1 of this paper and the only novelty of Algorithm 1 is the use of g=f-l (as opposed to f) as the utility function which leads to \alpha-regret bounds with curvature-dependent approximation ratio \alpha. - Since the multilinear extension of submodular set functions are defined as an expectation, many samples need to be used to obtain a good approximation for the gradient of the multilinear extension. I noticed that this issue has been addressed in more detail in the appendix, however, I think it's important to discuss the computational complexity of the discretized version of Algorithm 1 in the paper as well. In particular, implementing this algorithm on a real-world problem and comparing the performance for different discretization levels and various number of samples could have made the paper even better.

Correctness: All the claims of the paper have been well justified through mathematically sound arguments and proofs. The paper lacks numerical experiments.

Clarity: The paper is easy to read and even readers with limited background in submodularity and online learning would be able to follow the arguments and understand the techniques that have been used.

Relation to Prior Work: The authors have done a good job reviewing the previous literature on online submodular maximization, however, as it was mentioned earlier, a comparison of the algorithms of [9] and [10] with Algorithm 1 is missing. The algorithms seem to be almost identical and it is unclear what the novelty of Algorithm 1 exactly is.

Reproducibility: Yes

Additional Feedback: - The definition of the curvature of submodular functions in the footnote of page 2 is incorrect. - Would it be helpful to use the softmax extension (as opposed to the multilinear extension) of submodular set functions in the proposed algorithms? How does the regret bound change in that case? - What is the intuition for the first term (the benchmark) in the regret metric defined for the USM-balance subproblem in equation 4.1? - The distance function dist(.,.) used in Section 4.2 has not been defined in the paper.


Review 4

Summary and Contributions: The paper considers an online setting for submodular maximization. The setting is that at each timestep, the algorithm selections a set, the adversary reveals a function f_t, and the goal is to minimize the total regret. The authors provide efficient algorithms with improved regret bounds under several different regimes, when the function is monotone, nonmonotone and depending on the constraint.

Strengths: + The authors unify and improve current state-of-the-art regret bounds, improving on results from several earlier works. These include three previous works that studied this online model. + Technical contributions are described clearly, indicating which is novel to this work. These include: novel analysis of online cts. greedy and a novel use of Blackwell approachability. The techniques employed by the authors are both novel and sophisticated.

Weaknesses: - the authors do not list any applications of the online model they consider. It is not obvious to me what ML applications would fit under this model. Although previous works have studied this problem and presumably discuss applications, it would be nice to at least mention them here. - it is unclear what the time complexities of the algorithms are, although they are in polynomial time.

Correctness: Appears to be. But did not check completely.

Clarity: - The paper is well written.

Relation to Prior Work: - Context is clearly explained as discussed above.

Reproducibility: Yes

Additional Feedback: === after rebuttal === I remain positive about the paper.

[Author Response · NeurIPS 2020]

We thank all the reviewers for their time in reading our paper and providing thoughtful comments.

**Reviewer 1.**

• Thank you for pointing out the typo. We will fix this.

**Reviewer 2.**

• Regarding the adversary: we may assume that an online adaptive adversary where the adversary is allowed to
see the algorithm's coin flips but only after the algorithm has played its random set. This is because our analysis
never makes use of the fact that the functions are fixed by the adversary ahead of time (and similarly for online
dual averaging since it is a deterministic algorithm). We note that we obtain a guarantee in expectation.
• Regarding KL divergence: The KL divergence can be defined more generally for $x, y \in \mathbb{R}^n_{>0}$ as $d_{\mathrm{KL}}(x, y) =$
$\sum_{i=1}^n x_i \ln(x_i/y_i) - x_i + y_i$. It is not possible for any coordinate $y_i$ to be 0 as there is no point in the dual space
that may get mapped to a point with a non-positive component. One can also verify that the KL projection
does not cause any coordinate to become non-positive as well. We will add the details in the revised version.
• Regarding first-order regret bound: First-order regret bounds are a type of data-dependent bounds which often
depend on the magnitude of the costs. There are also "second-order" regret bounds which look at the "variance"
in the sequence of cost vectors.

**Reviewer 3.**

• Thanks for pointing out the typo in the definition of curvature. We will fix this.
• Regarding the regret bounds in [9, 10]: The previous work [9, 10] studies online continuous DR-submodular
maximization. Both regret bounds in [9, 10] involve the term $GD\sqrt{T}$, where $G$ is the upper bound of $\ell^2$-norm
of the gradients of objective functions and $D$ is the $\ell^2$-diameter of the feasible set. When applying their
algorithm to our setting (i.e., the objective functions are the multilinear extension and the feasible set is matroid
polytope), both $G$ and $D$ can be $\Omega(\sqrt{n})$, where $n$ is the size of the ground set. Hence, their bounds yields
$\Omega(n\sqrt{T})$, whereas our bound is $O(\sqrt{kT\log(n/k)})$. Even if we replace online gradient descent with mirror
descent in their algorithm, the regret bound is still (at least) $O(k\sqrt{T\log(n/k)})$. The improvement of factor
$\sqrt{k}$ in our algorithm comes from the use of the first order regret bound in OLO, which is the main contribution
of our paper. We will add the detailed comparison in the revised version.
• Regarding intuition for Equation 4.1: In the offline case, the proof of double greedy considers a potential
function of the form $2f(O_i) + f(X_i) + f(Y_i)$ where $O_i$ is OPT intersected with $Y_i$. The change in potential
at step $i$ of double greedy can be lower bounded by

$$- \max\{p_i^+ \Delta_i^-, p_i^- \Delta_i^+\} + \frac{1}{2}(p_i^+ \Delta_i^+ + p_i^- \Delta_i^-).$$

In offline double greedy, one chooses $p_i = (p_i^+, p_i^-)$ so that the above expression is non-negative, i.e. so that
the potential is non-decreasing. Equation 4.1 is the online equivalent of this change in potential (albeit with a
sign change).
• Sample complexity and time complexity:
– Monotone and matroid setting: we assume that the matroid is given by the rank oracle. Our algorithm
(Algorithm 6) makes $O(\frac{n^4}{\varepsilon^3} \log(\frac{n^3 T}{\varepsilon}))$ calls to the evaluation oracle of the objective function $f_t$ and solves
$O(\frac{n^3}{\varepsilon})$ submodular function minimization in each round $t$. Note that submodular function minimization
is used for the Bregman projection step.
– Nonmonotone and unconstrained setting: Our algorithm (Algorithm 2) makes $O(n)$ calls to the evaluation
oracle of the objective function $f_t$ and makes $O(n)$ overheads in each round $t$. Note that our USM-balance
subproblem algorithm runs in constant time since the underlying convex optimization is of constant
dimension.
• On softmax extension: The softmax extension can be efficiently computed for specific submodular functions
arising from determinant point processes, but it is unknown how to compute it for general submodular functions
faster than multilinear extension. Since our paper assumes the value oracle model, we did not use the softmax
extension. Note that the the sampling for evaluating the multilinear extension does not affect the regret bound.

**Reviewer 4.**

• On applications: Applications of online submodular maximization include learning blog rankings, online ad
display, online resource allocation. See [9, 10, 16, 29, 30] and references therein.
• On time complexity: please see the answer to Reviewer 3.

[Meta-Review · NeurIPS 2020]

A very solid theoretical result on online submodular maximization.